# PairFlow: Closed-Form Source-Target Coupling for Few-Step Generation in Discrete Flow Models

**Mingue Park**[*]    **Jisung Hwang**[*]    **Seungwoo Yoo**[*]    **Kyeongmin Yeo**    **Minhyuk Sung**

KAIST

`{kicikicik,4011hjs,dreamy1534,aaaaa,mhsung}@kaist.ac.kr`

## Abstract

We introduce PairFlow, a lightweight preprocessing step for training Discrete Flow Models (DFMs) to achieve few-step sampling without requiring a pretrained teacher. DFMs have recently emerged as a new class of generative models for discrete data, offering strong performance. However, they suffer from slow sampling due to their iterative nature. Existing acceleration methods largely depend on finetuning, which introduces substantial additional training overhead. PairFlow addresses this issue with a lightweight preprocessing step. Inspired by ReFlow and its extension to DFMs, we train DFMs from coupled samples of source and target distributions, without requiring any pretrained teacher. At the core of our approach is a closed-form inversion for DFMs, which allows efficient construction of paired source–target samples. Despite its extremely low cost, taking only up to 1.7% of the compute needed for full model training, PairFlow matches or even surpasses the performance of two-stage training involving finetuning. Furthermore, models trained with our framework provide stronger base models for subsequent distillation, yielding further acceleration after finetuning. Experiments on molecular data as well as binary and RGB images demonstrate the broad applicability and effectiveness of our approach. Project page: `https://pair-flow.github.io`

## 1 Introduction

Discrete Flow Models (DFMs) (Campbell et al., 2024; Gat et al., 2024) have recently emerged as a promising class of generative models, extending the idea of Flow Models (FMs) for continuous data to the discrete domain. By adapting flow-based principles to categorical structures, DFMs provide a principled and efficient way to capture complex discrete distributions through iterative sampling. They have shown success across a variety of applications, particularly in scientific domains such as molecule generation (Ramakrishnan et al., 2014; Irwin et al., 2012), where DFMs offer a natural framework for modeling chemical structures and generating novel candidates.

Analogous to FMs in the continuous domain, a key challenge of DFMs is the long computation time for generation due to their iterative sampling nature. Recent work (Deschenaux & Gulcehre, 2025; Hayakawa et al., 2025; Sahoo et al., 2025; Yoo et al., 2025) have sought to accelerate the generative process through distillation-based finetuning, which builds on ideas originally developed for continuous flow matching. Notably, ReFlow (Liu & Gong, 2023) is a well-known technique for FMs that pairs samples from the source (prior) distribution and the target distribution by running the generative process of a pretrained model and using the resulting pairs for finetuning. Recently, this idea has also been extended to DFMs (Yoo et al., 2025) to reduce conditional total correlation through finetuning, thereby enabling few-step generation.

Despite these promising results in acceleration, distillation-based methods incur substantial finetuning overhead, amounting to about 10–20% of the time required to train the base model from scratch. In other words, the gain in generation speed comes at the expense of considerable additional training cost. To our knowledge, no prior work has addressed this training-time cost when pursuing inference-time

---
[*]Equal contribution.

acceleration. This raises a natural question: can we achieve speedups comparable to distillation-based approaches while requiring only a lightweight preprocessing phase that requires orders of magnitude less compute, on the order of tens of GPU minutes?

We propose PAIRFLOW, a training framework for DFMs that enables few-step sampling by constructing paired source–target samples using closed-form velocities. **While inspired by ReDi-style coupling-driven training, our approach eliminates the need for a pretrained teacher by using closed-form formulations and achieves acceleration without finetuning.** The algorithm for computing source–target pairs is fully parallelizable and requires at most 1.7% of compute needed for full model training. Despite relying only on a lightweight preprocessing step, PAIRFLOW attains performance comparable to, or even superior to state-of-the-art distillation-based techniques, which can require up to 143 times more computation. Furthermore, models trained with our technique provide stronger bases for subsequent distillation, delivering additional performance gains while incurring only minimal preprocessing cost.

At the core of our framework is the simulation of probability paths connecting source (prior) and target (data) distributions in discrete spaces, made possible by closed-form expressions of velocities. While closed-form forward velocities have been studied for flow models in continuous domains (Karras et al., 2022; Bertrand et al., 2025), they have, to the best of our knowledge, neither been explored for DFMs nor applied to identifying suitable source–target pairs in the context of distillation-based acceleration, as in ReDi (Yoo et al., 2025). In this work, we investigate this idea for the first time. For DFMs, with a particular focus on uniform-state models (Sahoo et al., 2025; Schiff et al., 2025) equipped with a self-correcting mechanism, we show that the closed-form forward velocity is determined by the Hamming distance, which measures the number of differing tokens between two sequences. Using this velocity, samples from the source (latent) distribution can be mapped to given target samples. However, because multiple source samples may map to the same target, covering all targets through coupling would require an impractically large number of source samples. To overcome this, we derive the corresponding backward velocity in closed form and leverage it to simulate backward probability paths that efficiently map data points to source points, making pair discovery computationally efficient.

In our experiments, we show that the proposed framework enables few-step sampling across diverse discrete domains, including molecular data (Ramakrishnan et al., 2014; Irwin et al., 2012) and 2D images, exemplified by MNIST-Binary (LeCun et al., 2002) and CIFAR-10 (Krizhevsky et al., 2009). On the QM9 (Ramakrishnan et al., 2014) and ZINC-250k (Irwin et al., 2012) datasets, **PAIRFLOW not only improves the base model but also performs comparably to, or even better than, distilled models that require up to $143\times$ more compute during finetuning, compared to our lightweight preprocessing algorithm.** Similar improvements are observed on MNIST-Binary, where models paired with PAIRFLOW achieve performance comparable to those using DCD (Sahoo et al., 2025) and ReDi (Yoo et al., 2025), while being up to $35\times$ faster. Furthermore, after subsequent distillation, base models trained with pairs generated by our method consistently achieve higher performance, underscoring the importance of well-constructed source–target pairings.

## 2 RELATED WORK

### 2.1 DISCRETE FLOW MODELS

The concept of Flow Matching (Lipman et al., 2023) has recently been extended to the discrete flow-based models (Gat et al., 2024; Campbell et al., 2024; Sahoo et al., 2024; Schiff et al., 2025), demonstrating its flexibility across high-dimensional and structured data (Bai et al., 2025; Chang et al., 2022; Weber et al., 2024; Arriola et al., 2025; Nie et al.; Yu et al., 2023; Lee et al., 2025; Campbell et al., 2024; Wang et al., 2025). Among these, uniform-state models (Sahoo et al., 2025; Schiff et al., 2025) have been studied for their self-correcting properties, which enable recovery from errors introduced during parallel decoding. However, their performance degrades markedly in few-step settings, posing a key limitation for efficient generation under tight compute budgets.

### 2.2 ACCELERATING DISCRETE FLOW MODELS

Several approaches have been proposed to accelerate sampling with DFMs. Park et al. (2024) directly optimize sampling timesteps for improved parallelism while mitigating decoding errors. Hayakawa et al. (2025) highlight the importance of capturing dimensional correlations for faster sampling and

introduce mixture models to this end, at the cost of additional loss terms that complicate optimization. Another line of work adapts techniques from continuous domains, as in Sahoo et al. (2025), that propose a discrete analogue of Consistency Distillation (CD) (Song et al., 2023) by leveraging the duality between uniform-state and continuous Gaussian models. Most relevant to our approach is ReDi (Yoo et al., 2025), which draws inspiration from the concept of straight flows in ReFlow (Liu & Gong, 2023) and iteratively optimizes pairs of data and noise samples.

## 3 PRELIMINARIES

In this section, we provide a brief overview of flow matching for generative modeling of discrete data (Sec. 3.1), followed by a rectification technique (Yoo et al., 2025) that enables faster generation (Sec. 3.2) by reducing total correlation errors.

### 3.1 DISCRETE FLOW MATCHING

The goal of Discrete Flow Matching (DFM) (Campbell et al., 2024; Gat et al., 2024) is to learn a probability path $p_t(\cdot)$ that connects a known, easy-to-sample source distribution $p(x)$ to an unknown target distribution $q(x)$, both defined over a discrete state space. Once $p_t(\cdot)$ is known, samples from $q$ can be generated by drawing $x_0 \sim p$ and transporting it along the path.

Specifically, consider a sequence $x = (x^1, x^2, \ldots, x^N)$ of $N$ tokens, where each token takes values in a vocabulary $\mathcal{V} = \{1, 2, \ldots, K\}$ of size $K$. A sequence $x$ then resides in the product space $\mathcal{V}^N$. We denote by $\Delta^K = \{p \in \mathbb{R}^K \mid \sum_i p_i = 1, p_i \geq 0\}$ the probability simplex of dimension $K - 1$, on which distributions over $\mathcal{V}$ are defined.

Given a target probability path $p_t(x) : \mathcal{V}^N \times [0, 1] \to [0, 1]$ with an associated velocity field $v_t(x) : \mathcal{V}^N \times [0, 1] \to \mathbb{R}^{N \times K}$, we introduce a network $p_{1|t}^\theta(x) : \mathcal{V}^N \times [0, 1] \to (\Delta^K)^N$ to approximate $v_t(x)$. Its parameters $\theta$ are optimized via the DFM objective (Gat et al., 2024):

$$\mathcal{L}_{\text{DFM}}(\theta) = - \sum_{i \in \{1, \ldots, N\}} \mathbb{E}_{t \sim \mathcal{U}[0,1], x_0 \sim p, x_1 \sim q, z \sim p_t(x|x_0, x_1)} \log p_{1|t}^\theta(x_1^i | z), \tag{1}$$

where $p_{1|t}^\theta(x_1^i | z)$ denotes the learned probability denoiser, which predicts the categorical distribution of the clean token $x_1^i$ given an intermediate sequence $z$. Here, the conditional probability path $p_t(z | x_0, x_1)$ generates samples $z$ by interpolating between a data point $x_1 \sim q$ and a source sample $x_0 \sim p$. Assuming independence across tokens in sequence $x$, the conditional density factorizes as

$$p_t(z | x_0, x_1) = \prod_{i=1}^N p_t(z^i | x_0, x_1). \tag{2}$$

As token-wise conditional paths $p_t(z^i | x_0, x_1)$, Gat et al. (2024) employ the mixture path of form:

$$p_t(z^i | x_0, x_1) = (1 - \kappa_t) \delta_{x_0}(z^i) + \kappa_t \delta_{x_1}(z^i), \tag{3}$$

where the *scheduler* $\kappa_t = \kappa(t)$ is a monotonically increasing function over $t \in [0, 1]$ satisfying $\kappa_0 = 0$ and $\kappa_1 = 1$. For notational convenience, given $x, y \in \mathcal{V}^N$, we define the Dirac delta $\delta_y(x)$ as

$$\delta_y(x) = \prod_{i=1}^N \delta_{y^i}(x^i), \text{ where } \delta_{y^i}(x^i) = \begin{cases} 1 & x^i = y^i \\ 0 & x^i \neq y^i \end{cases}. \tag{4}$$

We also use the shorthand $\delta_y(x^i) = \delta_{y^i}(x^i)$. After optimizing $\theta$, the learned model parameterizes an approximation of the marginal velocity field:

$$v_t^\theta(x^i, z) = \frac{\dot{\kappa}_t}{1 - \kappa_t} \left[ p_{1|t}^\theta(x^i | z) - \delta_z(x^i) \right], \tag{5}$$

where $\dot{\kappa}_t = \frac{\partial \kappa_t}{\partial t}$. This learned velocity field $v_t^\theta(x^i, z)$ then transports samples over the interval $[0, 1]$ to simulate trajectories along $p_t(\cdot)$ and thereby generate samples. Each update step is defined as:

$$x_{t+h}^i \sim \text{Cat}\left(x_{t+h}^i; \delta_{x_t^i}(\cdot) + h \cdot v_t^\theta(x_{t+h}^i, x_t)\right), \tag{6}$$

where $h > 0$ is the step size.

## 3.2 STRAIGHTENING PROBABILITY PATHS FOR ACCELERATED SAMPLING

The concept of straight probability paths was originally introduced in the continuous domain to enable accelerated sampling. Prior work (Liu & Gong, 2023) identified curved probability paths as a key challenge in few-step sampling: when velocity fields are evaluated only at coarse time steps, numerical integration deviates from the true trajectories. Liu & Gong (2023) addressed this issue through "rectification," in which a student flow model is trained on source–target pairs generated by a teacher model, effectively yielding significantly straighter probability paths.

In the discrete setting, this challenge of *path curvature* translates to capturing the *statistical correlations* between tokens. Since DFMs approximate exponentially large joint transitions through fully factorized per-token updates, a mismatch inevitably arises between the true joint transition and its product-form approximation. This discrepancy becomes especially detrimental during few-step generation, where highly correlated tokens must be updated simultaneously. To address this, prior works have primarily relied on distillation-based approaches (Hayakawa et al., 2025; Sahoo et al., 2025; Deschenaux & Gulcehre, 2025), aiming to better capture these correlations by explicitly transferring multi-step dependencies from a stronger teacher model.

Yoo et al. (2025) formalized this factorization mismatch via conditional Total Correlation (TC), defined as:

$$\text{TC}_\pi(x_s|x_t) = \mathbb{E}_{x_t}\left[D_{\text{KL}}\left(p_{s|t}(x_s|x_t)\|\prod_{i=1}^{N}p_{s|t}(x_s^i|x_t)\right)\right],\tag{7}$$

which serves as a metric for the factorization error. Crucially, Yoo et al. (2025) interpret this factorization error as the discrete analog of path curvature: minimizing TC is equivalent to "straightening" the trajectory by decoupling token transitions. Analogous to ReFlow (Liu & Gong, 2023), which rectifies continuous paths, they demonstrate that reducing Eqn. 7 requires iteratively refining the source–target coupling $\pi(x_0, x_1)$. To achieve this, they employ an iterative distillation process, alternating between generating improved pairs using the current model and optimizing $\mathcal{L}_{\text{DFM}}$. This procedure effectively finds a "statistically straight" coupling that enables efficient few-step generation.

## 4 PAIRFLOW

For DFMs, ReDi (Yoo et al., 2025) improves sample quality in few-step generation by rectifying source-target pairs. However, it relies on samples from a pretrained model followed by costly retraining or finetuning. We take this one step further and pose the following question: What if these pairs could be generated directly from the data, without relying on a pretrained model or sampling from it?

To address this question, we propose a principled approach for discovering well-aligned source–target pairs without relying on pretrained models, enabling models trained on such pairs to achieve strong performance with few-step sampling. Our method, termed PAIRFLOW, leverages closed-form velocity fields that can be computed directly from the data samples, requiring only prior knowledge of the source distribution. We assume this distribution to be uniform, a choice extensively studied in recent work (Sahoo et al., 2025; Schiff et al., 2025), as models trained under this prior naturally acquire self-correcting properties.

In Sec. 4.1, we introduce the closed-form forward velocity for discrete flow matching (Gat et al., 2024). In Sec. 4.2, we extend this to the closed-form backward velocity and propose an algorithm for discovering well-aligned source–target pairs during the preprocessing phase.

## 4.1 FINDING PAIRS VIA CLOSED-FORM FORWARD VELOCITY FIELDS

As discussed in Sec. 3.1, DFMs (Campbell et al., 2024; Gat et al., 2024) aim to learn a marginal velocity field $v_t(\cdot)$ that induces a probability path $p_t(\cdot)$, transporting the source distribution $p_0(\cdot)$ to the target distribution $q(\cdot) = p_1(\cdot)$, which is unknown in practice. Instead, we only have access to a finite dataset of $M$ samples $\{d_m\}_{m=1}^{M}$. This empirical distribution $\tilde{q}(x)$ can be represented as a mixture of Dirac deltas centered at the observed samples:

$$q(x) \approx \tilde{q}(x) = \frac{1}{M}\sum_{m=1}^{M}\delta_{d_m}(x).\tag{8}$$

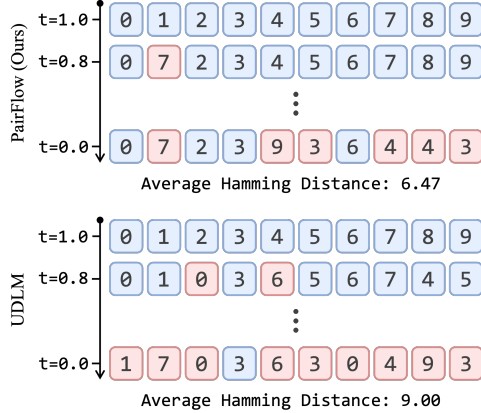

**Algorithm 1:** PAIRFLOW

1 **Input:** Dataset $\{d_m\}_{m=1}^M$, Steps $T$
2 **Output:** Pairs $\pi = \{(x_{0,m}, x_{1,m})\}_{m=1}^M$

3 Initialize $\pi \leftarrow \emptyset$
4 **for** $m \leftarrow 1$ **to** $M$ **do**
5     $x_{1,m} \leftarrow d_m$
6     $x \leftarrow x_{1,m}$
7     **for** $t \leftarrow 1$ **to** $T$ **do**
8        Compute $p_{0|t}(\cdot|x)$ and $\tilde{v}_t(\cdot, x)$ from Eqn. 12
9        Sample $z \sim \text{Cat}(z; \delta_x(\cdot) - h \cdot \tilde{v}_t(\cdot, x))$
10        $x \leftarrow z$
11     $x_{0,m} \leftarrow x$
12     $\pi \leftarrow \pi \cup \{(x_{0,m}, x_{1,m})\}$
13 **return** $\pi$

Figure 1: Illustrations of data inversion in PAIRFLOW and the standard corruption process in UDLM. PAIRFLOW achieves a lower average Hamming distance (6.47 vs. 9.0), promoting straighter paths during training.

For continuous domains, Karras et al. (2022); Bertrand et al. (2025) have shown that the velocity field transporting $p_0$ to $q$ can be derived in closed form when both distributions admit tractable density expressions. To the best of our knowledge, this idea has not been explored in discrete domains; in the following, we derive the closed-form velocity field for discrete domains for the first time.

We base our framework on the assumption of a uniform prior distribution over the discrete state space $\mathcal{V}^N$, defined as $p_0(x) = \mathcal{U}^N$, where $\mathcal{U} = \text{Cat}\left(\cdot; \frac{1}{K}\right)$ denotes the uniform distribution over the dictionary $\mathcal{V}$. For the empirical target distribution $\tilde{q}(x)$ introduced in Eqn. 8, we show in App. A.1 that the closed-form denoiser $p_{1|t}(x^i|z)$ and its associated velocity field $\hat{v}_t(x^i, z)$ are given by:

$$p_{1|t}(x^i|z) = \frac{\sum_{m=1}^M \delta_{d_m^i}(x^i)\gamma^{-h(d_m, z)}}{\sum_{m=1}^M \gamma^{-h(d_m, z)}} \implies \hat{v}_t(x^i, z) = \frac{\dot{\kappa}_t}{1 - \kappa_t}\left[p_{1|t}(x^i|z) - \delta_z(x^i)\right] \quad (9)$$

where $\gamma = (1 + (K-1)\kappa_t)/(1-\kappa_t)$, $K$ denotes the vocabulary size, and $h(s, z) = N - \sum_{i=1}^N \delta_{s^i}(z^i)$ is the Hamming distance between sequences $s$ and $z$, *i.e.,* the number of positions at which they differ. The token-wise denoiser $p_{1|t}(x^i|z)$ above is a weighted mixture of Dirac deltas, where sequences closer to $z$ under the Hamming distance contribute more. Intuitively, the forward velocity field $\hat{v}_t(x^i, z)$ pulls each token toward those from dataset sequences most similar to $z$. The most direct way to construct source-target pairs using $\hat{v}_t(x^i, z)$ is to sample $x_0 \sim p_0(x)$ and evolve it along the velocity field until it reaches a dataset point $x_1$. In practice, however, the generated data points fail to fully cover $\tilde{q}(x)$, requiring an impractically large number of source samples to achieve sufficient coverage. Our empirical results, reported in App. C.1, support this claim and motivate the exploration of a more efficient alternative, which we present in the following section.

## 4.2 FINDING PAIRS VIA CLOSED-FORM BACKWARD VELOCITY FIELDS

We address this issue by *backtracing* trajectories along $p_t(\cdot)$, starting from $\tilde{q}(x)$ and progressing toward the source distribution $p_0(\cdot)$. Unlike the forward construction in Sec. 4.1, this guarantees that all data points in $\tilde{q}(x)$ are included in the resulting pairs by design. As illustrated at the top of Fig. 1, PAIRFLOW inverts data samples toward the source distribution, assumed to be uniform. Unlike the standard corruption process used by UDLM (Schiff et al., 2025) shown at the bottom of Fig. 1, the source samples obtained by PAIRFLOW remain closer to the original data in terms of Hamming distance. This helps the model learn to recover data with fewer token transitions during training, effectively approximating the straight probability paths explored in ReFlow (Liu & Gong, 2023) and ReDi (Yoo et al., 2025).

Table 1: Dataset and training statistics. $N$ denotes the number of tokens per sample, $K$ the dictionary size, $|X_1|$ the dataset size, and $T_*$ the runtime of each method (in minutes, measured in wall-clock time with RTX A6000). The numbers in parentheses are the proportion of time relative to $T_{\text{Base}}$.

| Dataset | $N$ | $K$ | $|X_1|$ | $T_{\text{Base}}$ | $T_{\text{DCD}}$ | $T_{\text{ReDi}}$ | $T_{\text{PAIRFLOW}}$ |
|---|---|---|---|---|---|---|---|
| MNIST-Binary | 768 | 2 | 60,000 | 80 (100.0%) | 40 (50.0%) | 49 (61.0%) | 1.4 (1.7%) |
| CIFAR-10 | 3,072 | 256 | 100,000 | 6720 (100.0%) | 360 (5.3%) | 468 (6.9%) | 20 (0.3%) |
| QM9 | 32 | 40 | 127,190 | 450 (100.0%) | 115 (24.8%) | 100 (22.2%) | 0.8 (0.2%) |
| ZINC-250k | 72 | 74 | 224,568 | 1,110 (100.0%) | 211 (19.0%) | 194 (17.4%) | 13 (1.2%) |

The remaining challenge is to derive the closed-form backward velocity that governs this process. This can be obtained by following a construction analogous to Sec. 4.1. Specifically, we first derive the closed-form noise predictor $p_{0|t}(x^i|z)$:

$$p_{0|t}(x^i|z) = \delta_z(x^i) - \frac{\kappa_t(K\delta_{x^i}(z^i) - 1)}{1 + (K-1)\kappa_t} \frac{\sum_{m=1}^M \delta_{d_m^i}(z^i)\gamma^{-h(d_m,z)}}{\sum_{m=1}^M \gamma^{-h(d_m,z)}}, \tag{10}$$

with a detailed derivation provided in App. A.2. Substituting this into the definition of the backward velocity field from Gat et al. (2024)

$$\check{v}_t(x^i, z) = \frac{\dot{\kappa}_t}{\kappa_t}\left[\delta_z(x^i) - p_{0|t}(x^i|z)\right], \tag{11}$$

we obtain the desired closed-form expression

$$\check{v}_t(x^i, z) = \frac{\dot{\kappa}_t(K\delta_{x^i}(z^i) - 1)}{1 + (K-1)\kappa_t} \frac{\sum_{m=1}^M \delta_{d_m^i}(z^i)\gamma^{-h(d_m,z)}}{\sum_{m=1}^M \gamma^{-h(d_m,z)}}. \tag{12}$$

The second term in Eqn. 10 computes the conditional likelihood of the $i$-th token taking value $x^i \in \mathcal{V}$ given the current sequence $z$, marginalized over all dataset $\{d_m\}_{m=1}^M$. The contribution of each data sample $d_m$ is determined by its proximity to $z$ under the Hamming distance $h(d_m, z)$, assigning higher weight to tokens with greater local consensus. Consequently, updating with $\check{v}_t(x^i, z)$ pushes the sample away from the data distribution and toward the source distribution $p_0(x)$. Using $\check{v}_t(x^i, z)$, we construct pairs $\{(x_{0,m}, x_{1,m})\}_{m=1}^M$ by initializing from data points $\{d_m\}_{m=1}^M$ (equivalently, $\{x_{1,m}\}_{m=1}^M$) and iteratively applying the backward update rule in Eqn. 6 for a fixed number of iterations $T$. The overall procedure is summarized in Alg. 1.

## 5 EXPERIMENTAL RESULTS

We validate the effectiveness of the proposed method and the source–target pairs it discovers across several discrete generative modeling benchmarks involving molecular data and images. We first summarize the experimental setup in Sec. 5.1. In Sec. 5.2 and Sec. 5.3, we compare our method against baselines in molecular and image generation, respectively. In Sec. 5.4, we further demonstrate that models trained with pairs discovered by our method not only achieve improved performance directly, but also benefit subsequent distillation phases, as the resulting base model provides a stronger initialization for existing distillation techniques.

### 5.1 EXPERIMENT SETUP

**Baselines.** Across multiple benchmarks, we compare our approach against state-of-the-art discrete flow models, including MDLM (Sahoo et al., 2024) and UDLM (Schiff et al., 2025). Since our method is based on a uniform source distribution, we adopt UDLM (Schiff et al., 2025), the leading uniform-state model, as our base and denote UDLM trained with pairs generated by Alg. 1 as PAIRFLOW throughout the remainder of this section. In addition, we compare against these models augmented with distillation-based techniques that require additional finetuning, Discrete Consistency Distillation (DCD) (Sahoo et al., 2025) and ReDi (Yoo et al., 2025), denoted throughout this section by the suffixes "+ DCD" and "+ ReDi". The detailed training setups of these models, such as hyperparameters, are provided in App. B. Additionally, we report the performance of the same base model trained on pairs formed by each data point and a source sample randomly drawn from the uniform distribution with our detailed experimental results in App. D.

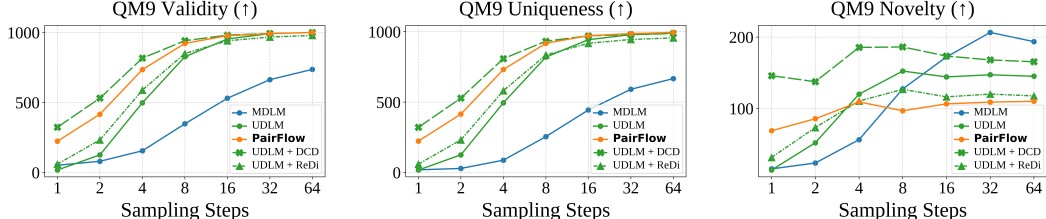

Figure 2: Step-wise performance analysis on the QM9 dataset (Ramakrishnan et al., 2014). Each plot reports the number of valid (left), unique (middle), and novel (right) SMILES strings (Weininger, 1988) out of 1,024 generated samples. Best viewed when zoomed in.

**Benchmarks.** We evaluate our method across a diverse set of discrete generative modeling benchmarks, covering both molecule and image generation tasks. For molecule generation, we experiment with the QM9 (Ramakrishnan et al., 2014) and ZINC-250k (Irwin et al., 2012) datasets. For image generation, we use the MNIST dataset (LeCun et al., 2002) with binarized pixel values (denoted MNIST-Binary) and the CIFAR-10 dataset (Krizhevsky et al., 2009), where pixel intensities are scaled to 8-bit integers, and horizontal flip augmentation is applied. Dataset statistics, including sample size, vocabulary size, and overall dataset size, are summarized in Tab. 1.

**Evaluation Setup.** For molecular generation, we follow Schiff et al. (2025) and evaluate the validity, uniqueness, and novelty of generated molecules. Specifically, we sample 1,024 SMILES strings (Weininger, 1988), convert them into molecular graphs, and compute these metrics. All results are averaged over 10 trials, with further details provided in App. D. For image generation, we report Fréchet Inception Distance (FID) (Heusel et al., 2017) and Inception Score (IS) (Salimans et al., 2016). FID is computed with 1,000 images for MNIST-Binary, and both FID and IS are computed with 5,000 generated images for CIFAR-10. The training dataset is used as the reference for FID computation. Across all experiments, we vary the number of sampling steps to evaluate performance in both low- and high-NFE settings. In particular, we generate samples using $1 - 64$ steps for molecular benchmarks (QM9 and ZINC-250k) and MNIST-Binary benchmark, and $8 - 1024$ steps for CIFAR-10 (Krizhevsky et al., 2009), as models yielded excessively high FIDs under extremely low-step settings.

## 5.2 MOLECULE GENERATION

We begin by benchmarking unconditional molecule generation, where models are tasked with generating SMILES strings (Weininger, 1988) that represent molecules. As illustrated in Fig. 2 and Fig. 3, which summarize validity (left), uniqueness (middle), and novelty (right), PAIRFLOW consistently improves upon its base model UDLM (Schiff et al., 2025), yielding substantial gains in few-step settings. It facilitates 1-step generation on QM9 (Ramakrishnan et al., 2014), a challenging setting that requires capturing all token-wise dependencies simultaneously. In this case, validity increases from 17.5 to 223.4, corresponding to a $12.8\times$ improvement. Similar trends are observed in the 2-step and 4-step settings, with validity improving by $231\%$ and $47.6\%$, respectively. As shown in Fig. 2 (left), this improvement is particularly significant: the 2-step and 4-step validities of PAIRFLOW are comparable to the 4-step and 8-step validities achieved by UDLM (Schiff et al., 2025). Comparable improvements are also seen on the ZINC-250k (Irwin et al., 2012) dataset.

Remarkably, **PAIRFLOW introduces minimal overhead—less than $2\%$ of the training cost as shown in Tab. 1—and requires no pretrained models, yet achieves performance comparable to, and in some cases surpassing, models distilled from the same base using DCD (Sahoo et al., 2025) and ReDi (Yoo et al., 2025)**, both of which rely on pretrained models and finetuning. On both QM9 (Ramakrishnan et al., 2014) and ZINC-250k (Irwin et al., 2012), PAIRFLOW consistently outperforms UDLM + ReDi across all few-step settings, achieving substantially higher 2-step validities on QM9 (232.4 vs. 416.0) and ZINC-250k (75.9 vs. 146.3). At the same time, PAIRFLOW matches the performance of UDLM + DCD, with comparable 2-step validities on QM9 (416.0 vs. 530.8). This is particularly notable given that the additional preprocessing cost of PAIRFLOW amounts to only $0.69\%$ on QM9 (Ramakrishnan et al., 2014) and $6.16\%$ on ZINC-250k (Irwin et al., 2012), relative to the full cost of DCD (Sahoo et al., 2025). Detailed numerical results with standard deviations are reported in App. D.

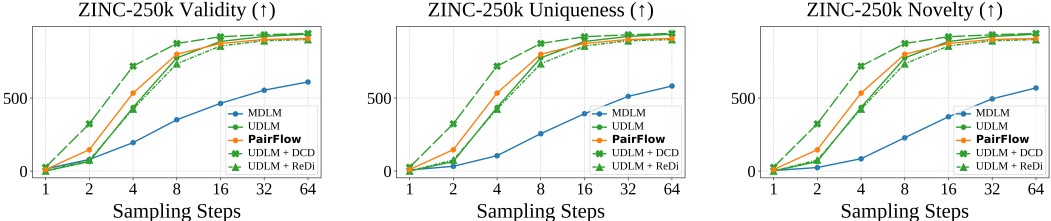

Figure 3: Step-wise performance analysis on the ZINC-250k dataset (Irwin et al., 2012). Each plot reports the number of valid (left), unique (middle), and novel (right) SMILES strings (Weininger, 1988) out of 1,024 generated samples. Best viewed when zoomed in.

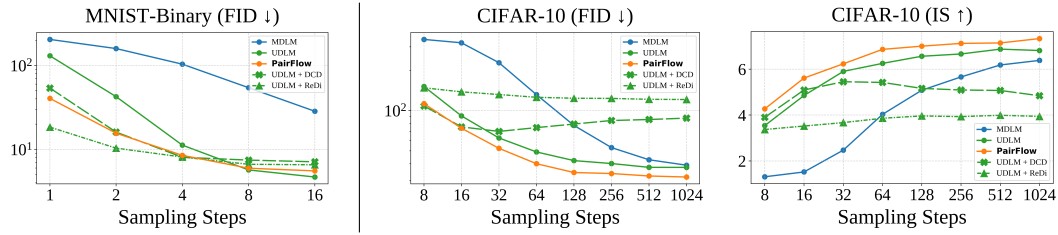

Figure 4: Step-wise performance analysis on discretized image datasets. From left to right: FID on MNIST-Binary (LeCun et al., 2002), FID on CIFAR-10 (Krizhevsky et al., 2009), and Inception Scores (IS) on CIFAR-10. Best viewed when zoomed in.

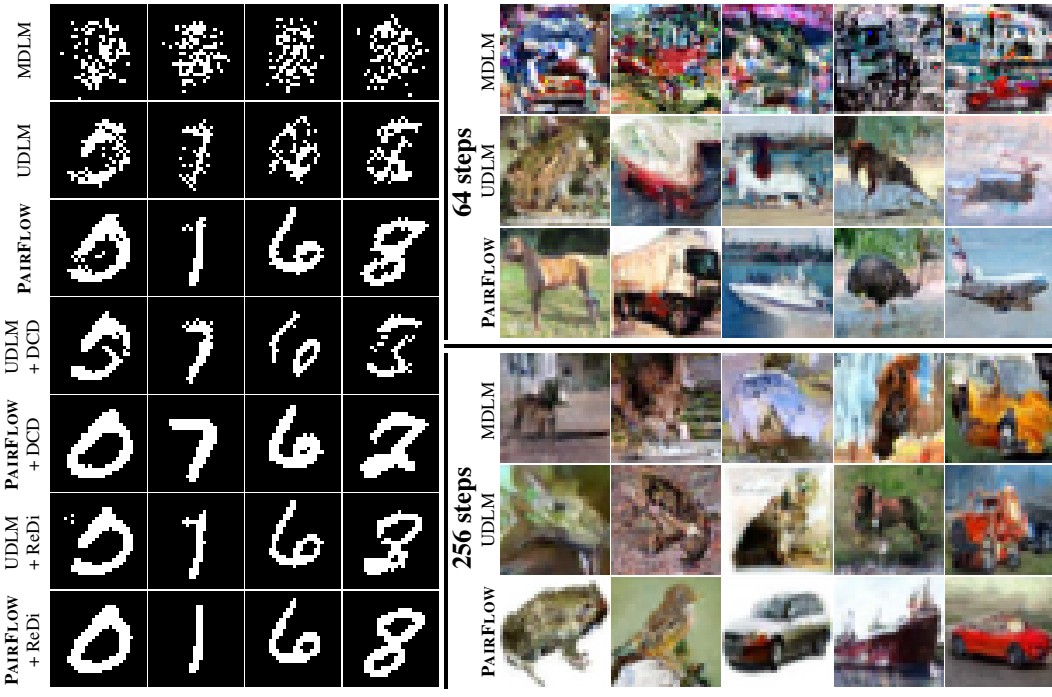

Figure 5: Qualitative results of 1-step generation on MNIST-Binary ($28 \times 28$; left) and 64-step (top right) and 256-step (bottom right) generation on CIFAR-10 ($32 \times 32$).

## 5.3 IMAGE GENERATION

We further extend our experiments to image domains where each pixel has discretized intensities. As in Sec. 5.2, we evaluate model performance across multiple sampling steps and summarize the results in Fig. 4. Qualitative samples for MNIST-Binary and CIFAR-10 are shown in Fig. 5. Both qualitative and quantitative results show that PAIRFLOW improves the performance of UDLM (Schiff et al., 2025) and, in few-step settings, achieves performance comparable to DCD (Sahoo et al., 2025).

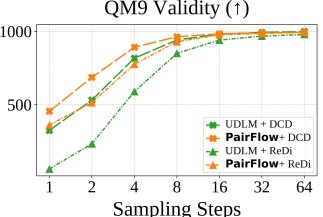 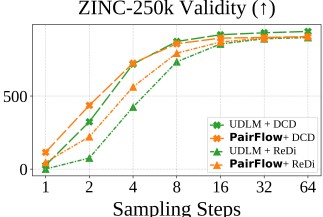 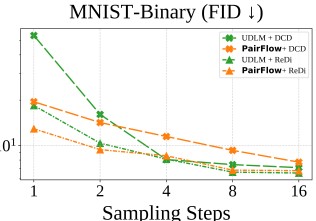

Figure 6: Step-wise performance analysis of distilled models on molecular and image datasets. From left to right: number of valid molecules on QM9, number of valid molecules on ZINC-250k, and FID on MNIST-Binary.

On MNIST-Binary (Fig. 4, left), PAIRFLOW achieves an FID of $40.59$ in the 1-step setting, equivalently a $68.9\%$ improvement over UDLM (Schiff et al., 2025). Consistent gains are observed across other few-step settings as well: at 2 steps, FID is reduced by $63.3\%$ ($15.61$ vs. $42.54$), and at 4 steps by $24.4\%$ ($8.51$ vs. $11.25$). On CIFAR-10 (Krizhevsky et al., 2009), where FID (Heusel et al., 2017) and IS (Salimans et al., 2016) are reported in Fig. 4 (middle and right), PAIRFLOW likewise outperforms the base UDLM, validating the effectiveness of the discovered source–target pairs.

As in molecular generation (Sec. 5.2), **PAIRFLOW performs comparable to distillation-based acceleration methods.** On MNIST-Binary, it achieves a lower FID than UDLM+DCD in the 1-step setting ($40.59$ vs. $53.84$) and comparable performance at 2 steps ($15.61$ vs. $16.09$). Likewise, PAIRFLOW performs competitively with UDLM+ReDi at 2 steps ($15.61$ vs. $10.36$), while requiring substantially less compute than both. As summarized in Tab. 1, DCD (Sahoo et al., 2025) requires 40 minutes ($T_{\mathrm{DCD}}$) and ReDi (Yoo et al., 2025) takes 49 minutes, whereas the preprocessing phase of PAIRFLOW completes in just $1.4$ minutes ($T_{\mathrm{PAIRFLOW}}$), yielding $28.6\times$ and $35\times$ speedups, respectively. On the CIFAR-10 benchmark, both DCD (Sahoo et al., 2025) and ReDi (Yoo et al., 2025) degrade model performance, as indicated by the higher FID in Fig. 4 (middle) and lower IS in Fig. 4 (right). The results in Tab. 13 suggest that, overall, acceleration methods do not work well on CIFAR-10. We hypothesize that this issue arises from the low performance of the teacher model, which negatively affects the student model when applying acceleration methods. Detailed results are reported in App. D.

## 5.4 DISTILLING MODELS TRAINED WITH ALIGNED PAIRS

While PAIRFLOW alone achieves performance comparable to, or even exceeding, distillation-based techniques (Yoo et al., 2025; Sahoo et al., 2025), as shown in Sec. 5.2 and Sec. 5.3, we further emphasize that it also serves as a strong initialization for subsequent distillation, yielding even greater performance gains when combined with existing methods. Crucially, this incurs negligible additional cost relative to the overall time required for distillation.

We validate this by distilling PAIRFLOW, trained on QM9 (Ramakrishnan et al., 2014), ZINC-250k (Irwin et al., 2012), and MNIST-Binary, using DCD (Sahoo et al., 2025) and ReDi (Yoo et al., 2025), and comparing their performance against distilled models whose teachers were the base UDLM (Schiff et al., 2025). As shown in Fig. 6, **student models distilled from PAIRFLOW, denoted PAIRFLOW+DCD and PAIRFLOW+ReDi, push the frontier of performance previously achieved by distillation-based techniques.** For example, on the QM9 dataset (Ramakrishnan et al., 2014), PAIRFLOW+DCD substantially improves validity over UDLM+DCD ($453.8$ vs. $323$ for 1 step, $685.8$ vs. $530.8$ for 2 steps). A similar trend is observed for PAIRFLOW+ReDi on ZINC-250k (Irwin et al., 2012), yielding higher scores in both 1-step ($46.3$ vs. $0.7$) and 2-step ($221.5$ vs. $75.9$) generation. Importantly, as summarized in Tab. 1, these gains are achieved at only minimal additional preprocessing cost: $3.15\%$ of the average runtime of distillation on MNIST-Binary, $0.77\%$ on QM9, and $6.42\%$ on ZINC-250k.

## 6 CONCLUSION

We have presented PAIRFLOW, a novel approach to accelerating the generative process of Discrete Flow Models (DFMs) through a lightweight preprocessing step performed prior to training. Our preprocessing, which couples source and target samples, requires only up to 1.7% of the base model training cost, making it at least 20× more efficient than finetuning, while still achieving comparable or even superior performance. The key enabler is the closed-form inversion, which eliminates the need for a pretrained teacher model.

**Acknowledgements.** This work was supported by IITP grants (RS-2024-00399817, RS-2025-25441313, RS-2025-25443318), funded by the Korean government (MSIT); the Industrial Technology Innovation Program (RS-2025-02317326), funded by the Korean government (MOTIE); the National Supercomputing Center (KSC-2025-CRE-0475); and the DRB-KAIST SketchTheFuture Research Center.

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

# A   PROOF FOR CLOSED-FORM VELOCITY

In this section, we present the detailed derivations of the closed-form forward velocity Eqn. 9 and backward velocity Eqn. 12 introduced in Sec. 4. Sec. A.1 provides the proof of the closed-form forward velocity, while Sec. A.2 presents the proof of the closed-form backward velocity. Both derivations are based on the assumption of a uniform source distribution.

## A.1   PROOF OF CLOSED-FORM FORWARD VELOCITY IN DISCRETE FLOW MODELS

Let $x, z \in \mathcal{V}^N$ be sequences of tokens $x^i, z^i \in \mathcal{V}$ for $i \in 1, \ldots, N$, where each token takes values from the discrete vocabulary $\mathcal{V} = \{1, \ldots, K\}$. We begin with the expression of the forward velocity given in Eqn. 9:

$$\hat{v}_t(x^i, z) = \frac{\dot{\kappa}_t}{1 - \kappa_t} \left[ p_{1|t}(x^i|z) - \delta_z(x^i) \right]. \tag{13}$$

We first derive the closed-form expression for the probability denoiser $p_{1|t}(x^i|z)$:

Using Bayes' rule,

$$p_{1|t}(x^i|z) = \sum_{x_0, x_1} \delta_{x_1^i}(x^i) \, p_t(x_0, x_1|z) \tag{14}$$

$$= \sum_{x_0, x_1} \delta_{x_1^i}(x^i) \, \frac{p_t(x_0, x_1, z)}{p_t(z)} \tag{15}$$

$$= \frac{\sum_{x_0, x_1} \delta_{x_1^i}(x^i) \, p_t(x_0, x_1, z)}{\sum_{x_0, x_1} p_t(x_0, x_1, z)}. \tag{16}$$

We factor the joint as

$$p_t(x_0, x_1, z) \propto p_0(x_0) \, p_1(x_1) \, p_t(z|x_0, x_1) \tag{17}$$

$$p_t(z|x_0, x_1) = \prod_{j=1}^{N} \left[ \kappa_t \, \delta_{x_1^j}(z^j) + (1 - \kappa_t) \, \delta_{x_0^j}(z^j) \right] \tag{18}$$

and use the empirical target

$$p_1(x_1) = \frac{1}{M} \sum_{m=1}^{M} \delta_{d_m}(x_1) \quad \text{(Eqn. 8)}. \tag{19}$$

Since $p_0(x_0)$ is constant, it cancels between the numerator and denominator of Eqn. 16, yielding

$$p_{1|t}(x^i|z) = \frac{\sum_{x_0, x_1} \delta_{x_1^i}(x^i) \, p_1(x_1) \, p_t(z|x_0, x_1)}{\sum_{x_0, x_1} p_1(x_1) \, p_t(z|x_0, x_1)}$$

$$= \frac{\sum_{m=1}^{M} \sum_{x_0} \delta_{d_m^i}(x^i) \prod_{j=1}^{N} \left[ \kappa_t \, \delta_{d_m^j}(z^j) + (1 - \kappa_t) \, \delta_{x_0^j}(z^j) \right]}{\sum_{m=1}^{M} \sum_{x_0} \prod_{j=1}^{N} \left[ \kappa_t \, \delta_{d_m^j}(z^j) + (1 - \kappa_t) \, \delta_{x_0^j}(z^j) \right]}. \tag{20}$$

According to this expression, the probability denoiser $p_{1|t}(x^i|z)$ can be interpreted as a weighted sum over all data points $x_1$, given by the term

$$\sum_{x_0} \prod_{j} \left[ \kappa_t \delta_{x_1^j}(z^j) + (1 - \kappa_t)\delta_{x_0^j}(z^j) \right]. \tag{21}$$

Let $h(s, z)$ denote the Hamming distance between two sequences $s$ and $z$, defined as

$$h(s, z) = N - \sum_{j} \delta_{s^j}(z^j), \tag{22}$$

and let $h_+(s, z)$ represent the similarity between the sequences, defined as

$$h_+(s, z) = \sum_j \delta_{s^j}(z^j) = N - h(s, z). \tag{23}$$

The weight is computed only when $d_m^i$ coincides with the target token $x^i$ (*i.e.*, $\delta_{d_m^i}(x^i) = 1$). In this case, the term can be expressed as:

$$\sum_{x_0} \prod_j \left[ \kappa_t \delta_{d_m^j}(z^j) + (1 - \kappa_t)\delta_{x_0^j}(z^j) \right] \tag{24}$$

$$= \sum_{k=0}^{h_+(d_m, z)} \binom{h_+(d_m, z)}{k} (1 - \kappa_t)^{N - h_+(d_m, z)} ((K - 1)\kappa_t)^{h_+(d_m, z) - k}. \tag{25}$$

To understand this transition, we first note that $d_m$ and $z$ are fixed in this scope, while $x_0$ is independent across each dimension and follows a uniform distribution. This implies that we only need to consider $x_0$. For an arbitrary dimension $j$, the cases can be divided into four possibilities, and the corresponding values of $\left[ \kappa_t \delta_{d_m^j}(z^j) + (1 - \kappa_t)\delta_{x_0^j}(z^j) \right]$ are as follows:

**Case 1.** $d_m^j = z^j$, $x_0^j = z^j$, $\left[ \kappa_t \delta_{d_m^j}(z^j) + (1 - \kappa_t)\delta_{x_0^j}(z^j) \right] = 1$.

**Case 2.** $d_m^j = z^j$, $x_0^j \neq z^j$, $\left[ \kappa_t \delta_{d_m^j}(z^j) + (1 - \kappa_t)\delta_{x_0^j}(z^j) \right] = \kappa_t$.

**Case 3.** $d_m^j \neq z^j$, $x_0^j = z^j$, $\left[ \kappa_t \delta_{d_m^j}(z^j) + (1 - \kappa_t)\delta_{x_0^j}(z^j) \right] = 1 - \kappa_t$.

**Case 4.** $d_m^j \neq z^j$, $x_0^j \neq z^j$, $\left[ \kappa_t \delta_{d_m^j}(z^j) + (1 - \kappa_t)\delta_{x_0^j}(z^j) \right] = 0$.

Note that **Case 4** makes the term inside the product $\left[ \kappa_t \delta_{d_m^j}(z^j) + (1 - \kappa_t)\delta_{x_0^j}(z^j) \right]$ equal to zero. Thus, we only need to consider $x_0$ for which no dimension falls into **Case 4**. Among the $|K|^N$ possible choices of $x_0$, only $|K|^{h_+(d_m, z)}$ satisfy $x_0^j = z^j$ for all dimensions where $d_m^j \neq z^j$. We then classify the remaining cases according to the Hamming distance between $x_0$ and $d_m$. Note that the maximum value of $h_+(x_0, d_m)$ is $h_+(d_m, z)$. Let $k$ denote an integer in the range 0 to $h_+(d_m, z)$. Then, the number of $x_0$ satisfying $h_+(x_0, d_m) = k$ is $\binom{h_+(d_m, z)}{k}(K - 1)^{h_+(d_m, z) - k}$, and in this case the product term becomes

$$\prod_j \left[ \kappa_t \delta_{d_m^j}(z^j) + (1 - \kappa_t)\delta_{x_0^j}(z^j) \right] = (\kappa_t)^{h_+(d_m, z) - k}(1 - \kappa_t)^{N - h_+(d_m, z)}. \tag{26}$$

We can then arrive at the equation above by summing over all possible $k$. Resuming the proof, the term can be further simplified as follows:

$$\sum_{x_0} \prod_j \left[ \kappa_t \delta_{d_m^j}(z^j) + (1 - \kappa_t)\delta_{x_0^j}(z^j) \right] \tag{27}$$

$$= \sum_{k=0}^{h_+(d_m, z)} \binom{h_+(d_m, z)}{k} (1 - \kappa_t)^{N - h_+(d_m, z)} ((K - 1)\kappa_t)^{h_+(d_m, z) - k} \tag{28}$$

$$= (1 - \kappa_t)^{N - h_+(d_m, z)} \sum_{k=0}^{h_+(d_m, z)} \binom{h_+(d_m, z)}{k} ((K - 1)\kappa_t)^k \tag{29}$$

$$= (1 - \kappa_t)^N \left( \frac{(K - 1)\kappa_t + 1}{1 - \kappa_t} \right)^{h_+(d_m, z)} \tag{30}$$

$$= (1 - \kappa_t)^N \left( 1 + \frac{\kappa_t}{1 - \kappa_t} K \right)^{h_+(d_m, z)}. \tag{31}$$

We define $\gamma := 1 + \frac{\kappa_t}{1-\kappa_t}K$, and by substituting this simplified expression into the noise predictor above, we finally obtain Eqn. 10.

$$p_{1|t}(x^i|z) = \frac{\sum_{m=1}^M \delta_{d_m^i}(x^i) \sum_{x_0} \prod_j \left[\kappa_t \delta_{d_m^j}(z^j) + (1-\kappa_t)\delta_{x_0^j}(z^j)\right]}{\sum_{m=1}^M \sum_{x_0} \prod_j \left[\kappa_t \delta_{d_m^j}(z^j) + (1-\kappa_t)\delta_{x_0^j}(z^j)\right]} \tag{32}$$

$$= \frac{\sum_{m=1}^M \delta_{d_m^i}(x^i)(1-\kappa_t)^N \left(1 + \frac{\kappa_t}{1-\kappa_t}K\right)^{h_+(d_m,z)}}{\sum_{m=1}^M (1-\kappa_t)^N \left(1 + \frac{\kappa_t}{1-\kappa_t}K\right)^{h_+(d_m,z)}} \tag{33}$$

$$= \frac{\sum_{m=1}^M \delta_{d_m^i}(x^i) \left(1 + \frac{\kappa_t}{1-\kappa_t}K\right)^{h_+(d_m,z)}}{\sum_{m=1}^M \left(1 + \frac{\kappa_t}{1-\kappa_t}K\right)^{h_+(d_m,z)}} \tag{34}$$

$$= \frac{\sum_{m=1}^M \delta_{d_m^i}(x^i)\gamma^{-h(d_m,z)}}{\sum_{m=1}^M \gamma^{-h(d_m,z)}}. \tag{35}$$

## A.2 Proof of Closed-form Backward Velocity in Discrete Flow Models

Similarly to the proof of the closed-form forward velocity in Sec. A.1, we start from the backward velocity in Eqn. 11:

$$\check{v}_t(x^i, z) = \frac{\dot{\kappa}_t}{\kappa_t}\left[\delta_{z^i}(x^i) - p_{0|t}(x^i|z)\right]. \tag{36}$$

We derive the closed-form noise predictor as follows:

$$p_{0|t}(x^i|z) = \sum_{x_0,x_1} \delta_{x_0^i}(x^i)p_t(x_0,x_1|z) \tag{37}$$

$$= \sum_{x_0,x_1} \delta_{x_0^i}(x^i)\frac{p_t(x_0,x_1,z)}{p_t(z)} \tag{38}$$

$$= \frac{\sum_{x_0,x_1} \delta_{x_0^i}(x^i)p_t(x_0,x_1,z)}{p_t(z)} \tag{39}$$

$$= \frac{\sum_{x_0,x_1} \delta_{x_0^i}(x^i)p_t(x_0,x_1,z)}{\sum_{x_0,x_1} p_t(x_0,x_1,z)}, \tag{40}$$

The last expression is further expanded to:

$$p_{0|t}(x^i|z) = \frac{\sum_{x_0,x_1} \delta_{x_0^i}(x^i)p_t(x_0,x_1,z)}{\sum_{x_0,x_1} p_t(x_0,x_1,z)} \tag{41}$$

$$= \frac{\sum_{x_0,x_1} \delta_{x_0^i}(x^i)\,p_1(x_1)\,p_t(z|x_0,x_1)}{\sum_{x_0,x_1} p_1(x_1)\,p_t(z|x_0,x_1)} \tag{42}$$

$$= \frac{\sum_{m=1}^M \sum_{x_0} \delta_{x_0^i}(x^i) \prod_{j=1}^N \left[\kappa_t\,\delta_{d_m^j}(z^j) + (1-\kappa_t)\,\delta_{x_0^j}(z^j)\right]}{\sum_{m=1}^M \sum_{x_0} \prod_{j=1}^N \left[\kappa_t\,\delta_{d_m^j}(z^j) + (1-\kappa_t)\,\delta_{x_0^j}(z^j)\right]}. \tag{43}$$

For the denominator, we use the same formula as in Eqn. 31:

$$\sum_{m=1}^M \sum_{x_0} \prod_{j=1}^N \left[\kappa_t\delta_{d_m^j}(z^j) + (1-\kappa_t)\delta_{x_0^j}(z^j)\right] = \sum_{m=1}^M (1-\kappa_t)^N \left(1 + \frac{\kappa_t}{1-\kappa_t}K\right)^{h_+(d_m,z)}. \tag{44}$$

Next, for the numerator, we can rewrite it as:

$$\sum_{m=1}^{M} \sum_{x_0} \delta_{x_0^i}(x^i) \prod_{j=1}^{N} \left[ \kappa_t \, \delta_{d_m^j}(z^j) + (1 - \kappa_t) \, \delta_{x_0^j}(z^j) \right] \tag{45}$$

$$= \sum_{m=1}^{M} \sum_{\substack{x_0 \text{ with} \\ x_0^i = x^i}} \prod_{j} \left[ \kappa_t \delta_{d_m^j}(z^j) + (1 - \kappa_t) \delta_{x_0^j}(z^j) \right]. \tag{46}$$

The $i$-th index should be considered separately as $x_0^i$ is set to be equal to $x^i$. Separating the $j = i$ term from the product yields

$$\sum_{m=1}^{M} \sum_{\substack{x_0 \text{ with} \\ x_0^i = x^i}} \left[ \kappa_t \delta_{d_m^i}(z^i) + (1 - \kappa_t) \delta_{x_0^i}(z^i) \right] \prod_{j \neq i} \left[ \kappa_t \delta_{d_m^j}(z^j) + (1 - \kappa_t) \delta_{x_0^j}(z^j) \right] \tag{47}$$

$$= \sum_{m=1}^{M} \sum_{\substack{x_0 \text{ with} \\ x_0^i = x^i}} \left[ \kappa_t \delta_{d_m^i}(z^i) + (1 - \kappa_t) \delta_{x^i}(z^i) \right] \prod_{j \neq i} \left[ \kappa_t \delta_{d_m^j}(z^j) + (1 - \kappa_t) \delta_{x_0^j}(z^j) \right] \tag{48}$$

$$= \sum_{m=1}^{M} \left[ \kappa_t \delta_{d_m^i}(z^i) + (1 - \kappa_t) \delta_{x^i}(z^i) \right] \sum_{\substack{x_0 \text{ with } j \neq i \\ x_0^i = x^i}} \prod \left[ \kappa_t \delta_{d_m^j}(z^j) + (1 - \kappa_t) \delta_{x_0^j}(z^j) \right]. \tag{49}$$

Since the $i$-th coordinate of $x_0$ is fixed to $x^i$, the summation over $x_0$ with $x_0^i = x^i$ no longer depends on this index. Consequently, when we consider the summation only over the remaining coordinates $j \neq i$, the resulting expression takes exactly the same form as the computation presented in Sec. A.1 (Eqn. 27-Eqn. 31). The only differences are that (i) the effective dimensionality of the product is reduced from $N$ to $N - 1$, and (ii) the matching count term must exclude the $i$-th coordinate, yielding $h_+(d_m, z)$ to $h_+(d_m, z) - \delta_{d_m^i}(z^i)$. Reflecting these adjustments, we obtain

$$\sum_{\substack{x_0 \text{ with } j \neq i \\ x_0^i = x^i}} \prod \left[ \kappa_t \delta_{d_m^j}(z^j) + (1 - \kappa_t) \delta_{x_0^j}(z^j) \right] = (1 - \kappa_t)^{N-1} \left( 1 + \frac{\kappa_t}{1 - \kappa_t} K \right)^{h_+(d_m, z) - \delta_{d_m^i}(z^i)},$$
$$\tag{50}$$

and

$$\sum_{m=1}^{M} \left[ \kappa_t \delta_{d_m^i}(z^i) + (1 - \kappa_t) \delta_{x^i}(z^i) \right] \sum_{\substack{x_0 \text{ with } j \neq i \\ x_0^i = x^i}} \prod \left[ \kappa_t \delta_{d_m^j}(z^j) + (1 - \kappa_t) \delta_{x_0^j}(z^j) \right] \tag{51}$$

$$= \sum_{m=1}^{M} \left[ \kappa_t \delta_{d_m^i}(z^i) + (1 - \kappa_t) \delta_{x^i}(z^i) \right] (1 - \kappa_t)^{N-1} \left( 1 + \frac{\kappa_t}{1 - \kappa_t} K \right)^{h_+(d_m, z) - \delta_{d_m^i}(z^i)} \tag{52}$$

$$= \sum_{m=1}^{M} \left[ \frac{\kappa_t}{1 - \kappa_t} \delta_{d_m^i}(z^i) + \delta_{x^i}(z^i) \right] (1 - \kappa_t)^{N} \left( 1 + \frac{\kappa_t}{1 - \kappa_t} K \right)^{h_+(d_m, z) - \delta_{d_m^i}(z^i)}. \tag{53}$$

Using this nominator, we can denote the closed-form noise predictor:

$$p_{0|t}(x^i|z) = \frac{\sum_{m=1}^{M}\left[\frac{\kappa_t}{1-\kappa_t}\delta_{d_m^i}(z^i) + \delta_{x^i}(z^i)\right](1-\kappa_t)^N\left(1+\frac{\kappa_t}{1-\kappa_t}K\right)^{h_+(d_m,z)-\delta_{d_m^i}(z^i)}}{\sum_{m=1}^{M}(1-\kappa_t)^N\left(1+\frac{\kappa_t}{1-\kappa_t}K\right)^{h_+(d_m,z)}} \tag{54}$$

$$= \sum_{m=1}^{M}\left[\frac{\kappa_t}{1-\kappa_t}\delta_{d_m^i}(z^i) + \delta_{x^i}(z^i)\right]\frac{\left(1+\frac{\kappa_t}{1-\kappa_t}K\right)^{h_+(d_m,z)-\delta_{d_m^i}(z^i)}}{\sum_{m'=1}^{M}\left(1+\frac{\kappa_t}{1-\kappa_t}K\right)^{h_+(d_{m'},z)}} \tag{55}$$

$$= \sum_{m=1}^{M}\left[\frac{\kappa_t}{1-\kappa_t}\delta_{d_m^i}(z^i) + \delta_{x^i}(z^i)\right]\left[1 - \frac{K\kappa_t\,\delta_{d_m^i}(z^i)}{1+(K-1)\kappa_t}\right]\frac{\gamma^{h_+(d_m,z)}}{\sum_{m'=1}^{M}\gamma^{h_+(d_{m'},z)}} \tag{56}$$

$$= \delta_{x^i}(z^i) - \frac{\kappa_t\left(K\delta_{x^i}(z^i)-1\right)}{1+(K-1)\kappa_t}\sum_{m=1}^{M}\delta_{d_m^i}(z^i)\frac{\gamma^{h_+(d_m,z)}}{\sum_{m'=1}^{M}\gamma^{h_+(d_{m'},z)}}, \tag{57}$$

where $\gamma := 1 + \frac{\kappa_t}{1-\kappa_t}K$.

We can then express the closed-form backward velocity as follows:

$$\check{v}_t(x^i,z) = \frac{\dot{\kappa}_t}{\kappa_t}\left[\delta_{z^i}(x^i) - p_{0|t}(x^i|z)\right] \tag{58}$$

$$= \frac{\dot{\kappa}_t\left(K\delta_{x^i}(z^i)-1\right)}{1+(K-1)\kappa_t}\sum_{m=1}^{M}\delta_{d_m^i}(z^i)\frac{\gamma^{h_+(d_m,z)}}{\sum_{m'=1}^{M}\gamma^{h_+(d_{m'},z)}} \tag{59}$$

$$= \frac{\dot{\kappa}_t(K\delta_{x^i}(z^i)-1)}{1+(K-1)\kappa_t}\sum_{m=1}^{M}\delta_{d_m^i}(z^i)\frac{\gamma^{-h(d_m,z)}}{\sum_{m'=1}^{M}\gamma^{-h(d_{m'},z)}}. \tag{60}$$

Since $\kappa_t = 1$ for $t = 1$, then $\gamma \to \infty$. So this equation is formally is not defined at $t = 1$. Nevertheless, as $\lim_{t\to1}$, the weighted sum over power of $\gamma$ is dominated by the maximum term, which converges to 1. Hence, the expression can be rigorously interpreted as $\lim_{t\to1}\check{v}_t(x^i,z)$, and in practice, this limiting value is used for sampling at $t = 1$.

## B  EXPERIMENT DETAILS

In Tab. 2, we summarize the hyperparameters used in the experiments presented in Sec. 5, covering both training and finetuning configurations for each dataset. All reported samples were generated using the greedy-tail denoiser described in (Sahoo et al., 2025). We employed an implementation of the closed-form backward velocity that is optimized at the CUDA level.

For the CIFAR-10 dataset, we follow the same setting as baseline (Schiff et al., 2025). Tab. 3 reports the FID and IS of baseline and PAIRFLOW measured with 1,000 steps and 50K samples, which are consistent with the results reported in Tab. 6 of (Schiff et al., 2025) and PAIRFLOW outperforms it.

## C  ADDITIONAL EXPERIMENTS

### C.1  COVERAGE OF TRAINING DATASET BY SAMPLING WITH FORWARD VELOCITY

As discussed in Sec. 4.1, constructing pairs using the closed-form forward velocity with a training dataset of size $|X_1|$ incurs significantly higher cost to achieve full coverage compared to using the backward velocity. Let $k$ denote the number of source samples drawn from the source (prior) distribution, assumed to be uniform in our work. The probability that a given element in the training set is selected is $\left(1 - \frac{1}{|X_1|}\right)^k$. Accordingly, we denote by $\bar{k}$ the number of unique samples among the $k$ draws, whose expectation is: $\sum_{i=1}^{|X_1|}\left[1 - (1 - \frac{1}{|X_1|})^k\right] = |X_1|\left[1 - (1 - \frac{1}{|X_1|})^k\right]$. In addition,

Table 2: Summary of the training settings used in Sec. 5. Specifically, "Sampling Steps" under PAIR-FLOW and ReDi (Yoo et al., 2025) indicate the number of steps taken to generate pairs, "Teacher Update Period" under DCD (Sahoo et al., 2025) denotes the number of fine-tuning iterations between updates, when the teacher model is replaced by the current student model. "# Pairs" under ReDi (Yoo et al., 2025) denotes the number of pairs for the fine-tuning.

| | MNIST-Binary | QM9 | CIFAR10 | ZINC-250k |
|---|---|---|---|---|
| Training Iterations | 10K | 50K | 300K | 200K |
| Data Dimension | $28 \times 28$ | 32 | $32 \times 32 \times 3$ | 72 |
| Batch Size | 256 | 1024 | 512 | 256 |
| Network Architecture | U-Net | Transformer | U-Net | Transformer |
| Parameter Count | 25.8M | 92.4M | 35.8M | 92.4M |
| | PAIRFLOW | | | |
| Sampling Steps | 20 | 20 | 20 | 64 |
| | DCD (Sahoo et al., 2025) | | | |
| Training Iterations | 5K | 10K | 50K | 30K |
| Teacher Update Period | 1K | 2K | 10K | 5K |
| | ReDi (Yoo et al., 2025) | | | |
| Training Iterations | 5K | 10K | 50K | 30K |
| # Pairs | 10K | 20K | 10K | 20K |
| Sampling Steps | 256 | 256 | 1024 | 256 |

Table 3: FID (Heusel et al., 2017) and IS (Salimans et al., 2016) of UDLM (Schiff et al., 2025) and PAIRFLOW on the CIFAR-10 dataset (Krizhevsky et al., 2009).

| | FID | IS |
|---|---|---|
| UDLM | 33.65 | 6.96 |
| PAIRFLOW | **28.07** | **7.37** |

Table 4: Total Correlation measure with pairs sampled from UDLM (Schiff et al., 2025) and PAIRFLOW trained on QM9 (Ramakrishnan et al., 2014).

| | Total Correlation |
|---|---|
| UDLM | 31.87 |
| PAIRFLOW | **30.72** |

Table 5: Summary of training set sizes $|X_1|$ for each dataset, the number of unique samples $\bar{k}$ obtained by simulating paths using the closed-form forward velocity in Eqn. 9, and the corresponding coverage values: empirical (COV) and theoretically predicted (COV$_{\text{Pred}}$).

| | QM9 | ZINC-250k | MNIST-Binary | CIFAR-10 |
|---|---|---|---|---|
| $|X_1|$ | 127,190 | 224,568 | 60,000 | 100,000 |
| $\bar{k}$ | 77,104 | 140,779 | 37,711 | 63,117 |
| COV | 60.62% | 62.68% | 62.85% | 63.11% |
| COV$_{\text{Pred}}$ | | 63.21% | | |

we define the coverage as the ratio between the number of unique elements obtained through this sampling procedure and the training set size: COV $= \bar{k}/|X_1|$.

To validate our claim in Sec. 4.1, we sample $k = |X_1|$ data points $x_1$ by transporting source samples $x_0$, independently drawn from the uniform distribution, along the velocity field defined in Eqn. 9. Using these samples, we evaluate the coverage following the definition above. The dataset sizes, number of unique samples among the generated samples, and the empirical and theoretical coverages are summarized in Tab. 5. These findings indicate that, even when sampling the same number of points as the training set size, only about 63% of the training distribution can be recovered in practice. Achieving full coverage would therefore require a substantially larger number of samples, introducing significantly higher computational cost. Motivated by this finding, we instead propose tracing backward from data samples, using a closed-form velocity field that we derive for this purpose (Sec. 4.2).

## C.2 TOTAL CORRELATION ANALYSIS OF CLOSED-FORM VELOCITY

As in Sec. 3.2, Yoo et al. (2025) demonstrated that iteratively refining the joint distribution of source-target pairs in discrete flow models improves few-step performance by reducing the total correlation of the model. In this section, we measure the total correlation following their methodology. Specifically, we perform sampling with neural networks, including UDLM (Schiff et al., 2025) and PAIRFLOW trained on QM9 (Ramakrishnan et al., 2014), starting from identical initial states $x_0$ but with varying random seeds. We randomly select 20,000 initial states $x_0$, and for each $x_0$, we generate 10 samples with a step size of 256. As shown in Tab. 4, PAIRFLOW achieves a lower total correlation, consistent with the improved performance observed in few-step sampling, as discussed above.

## D DETAILED EXPERIMENTAL RESULTS

In this section, we provide the detailed experimental results corresponding to those summarized in Sec. 5. For the molecular datasets (QM9 (Ramakrishnan et al., 2014) and ZINC-250k (Irwin et al., 2012)), we generate 1,024 samples across varying timesteps and evaluate validity, uniqueness, and novelty. Reported values are averaged over 10 trials, with standard deviations also included. For the image domain, we report FID on MNIST-Binary (LeCun et al., 2002), and both FID (Heusel et al., 2017) and IS (Salimans et al., 2016) on CIFAR-10 (Krizhevsky et al., 2009). Detailed experimental settings are provided in Sec. 5.1.

Results on QM9 are presented in Tables 6, 7, and 8, reporting validity, uniqueness, and novelty, respectively. Corresponding results on ZINC-250k are shown in Tables 9, 10, and 11. Finally, results for the image datasets are summarized in Tab. 12 (FID on MNIST-Binary) and Tab. 13 (FID and IS on CIFAR-10). The FID measured on MNIST-Binary (LeCun et al., 2002), FID and IS measured on CIFAR-10 (Krizhevsky et al., 2009), are summarized in Tab. 12 and Tab. 13, respectively.

Table 6: Validity scores (↑) on QM9 (Ramakrishnan et al., 2014) for various methods across different steps. Best values per column are highlighted in bold.

| Method | 1 | 2 | 4 | 8 | 16 | 32 | 64 |
|---|---|---|---|---|---|---|---|
| MDLM | 51.4±5.7 | 80.0±11.1 | 154.8±10.0 | 347.5±11.5 | 530.0±11.6 | 662.9±16.5 | 736.4±16.4 |
| UDLM | 17.5±3.2 | 125.5±11.8 | 497.6±8.3 | 826.6±10.3 | 953.5±6.1 | **991.9**±4.2 | 1000.1±3.5 |
| Random | 47.1±5.7 | 194.6±8.2 | 554.2±13.3 | 858.3±17.5 | 962.0±6.7 | 989.9±7.6 | 998.6±5.3 |
| PAIRFLOW | **223.4**±12.7 | **416.0**±12.4 | **734.9**±7.2 | **921.5**±11.0 | **977.1**±3.9 | 990.9±5.9 | **1000.2**±4.5 |
| UDLM + DCD | 323.0±19.5 | 530.8±20.0 | 816.6±14.4 | 941.1±8.5 | 981.0±4.8 | **993.0**±3.6 | **999.4**±4.7 |
| PAIRFLOW + DCD | **453.8**±16.4 | **685.8**±16.7 | **891.6**±11.9 | **963.1**±7.8 | **983.7**±8.5 | 989.3±3.5 | 993.2±5.7 |
| UDLM + ReDi | 59.7±8.8 | 232.4±9.2 | 588.4±15.8 | 849.6±14.2 | 940.5±8.5 | 967.5±5.2 | 978.8±5.2 |
| PAIRFLOW + ReDi | **361.0**±115.2 | **512.6**±44.2 | **775.7**±10.0 | **929.1**±11.6 | **976.2**±4.5 | **985.6**±6.7 | **993.1**±7.1 |

Table 7: Uniqueness scores (↑) on QM9 (Ramakrishnan et al., 2014) for various methods across different steps. Best values per column are highlighted in bold.

| Method | 1 | 2 | 4 | 8 | 16 | 32 | 64 |
|---|---|---|---|---|---|---|---|
| MDLM | 18.9±3.0 | 28.8±3.3 | 87.4±8.4 | 254.7±10.3 | 443.8±16.4 | 591.0±18.6 | 666.9±17.5 |
| UDLM | 17.5±3.2 | 125.4±11.7 | 495.0±8.2 | 819.5±11.4 | 943.0±5.7 | 979.1±5.0 | 990.0±4.7 |
| Random | 47.1±5.7 | 194.5±8.1 | 551.2±12.5 | 853.1±16.9 | 953.5±7.8 | 981.1±6.7 | 989.6±5.7 |
| PAIRFLOW | **223.0**±12.3 | **414.7**±12.0 | **731.4**±6.9 | **917.4**±11.8 | **971.6**±4.3 | **986.2**±5.3 | **994.8**±5.2 |
| UDLM + DCD | 320.9±18.7 | 528.0±19.7 | 808.5±12.7 | 932.3±8.1 | 970.9±5.8 | 978.3±4.9 | 987.2±4.3 |
| PAIRFLOW + DCD | **451.8**±15.7 | **681.6**±16.5 | **886.5**±12.6 | **957.8**±7.6 | **978.7**±9.1 | **983.4**±3.9 | **989.0**±5.3 |
| UDLM + ReDi | 59.7±8.8 | 231.6±9.5 | 581.0±15.1 | 834.7±11.4 | 917.5±9.6 | 944.8±6.2 | 956.1±5.2 |
| PAIRFLOW + ReDi | **359.5**±113.1 | **507.7**±43.0 | **765.3**±8.8 | **913.5**±10.2 | **959.7**±5.8 | **968.8**±8.4 | **973.0**±9.6 |

Table 8: Novelty scores (↑) on QM9 (Ramakrishnan et al., 2014) for various methods across different steps. Best values per column are highlighted in bold.

| Method | 1 | 2 | 4 | 8 | 16 | 32 | 64 |
|---|---|---|---|---|---|---|---|
| MDLM | 15.4±3.1 | 23.6±2.6 | 56.3±5.2 | 127.4±9.4 | **172.2**±12.4 | **206.5**±7.4 | **193.6**±10.4 |
| UDLM | 13.8±2.9 | 52.0±8.2 | 120.0±3.8 | 152.4±9.1 | 144.2±12.4 | 147.2±9.7 | 145.1±9.0 |
| Random | 26.9±5.0 | 66.5±7.1 | **128.8**±13.0 | **156.8**±10.6 | 145.5±9.1 | 146.5±7.0 | 150.1±8.7 |
| PAIRFLOW | **68.8**±7.8 | **85.6**±10.0 | 109.2±7.8 | 96.8±9.9 | 106.5±12.5 | 108.9±9.4 | 110.0±9.9 |
| UDLM + DCD | **145.9**±7.9 | **137.5**±9.8 | **185.5**±10.9 | **186.2**±13.7 | **173.4**±15.5 | **168.0**±10.4 | **165.4**±11.6 |
| PAIRFLOW + DCD | 110.3±6.3 | 136.1±12.0 | 146.2±12.9 | 139.2±9.6 | 131.8±11.7 | 140.1±9.6 | 133.2±7.9 |
| UDLM + ReDi | 31.4±8.0 | 73.3±7.1 | **110.4**±12.2 | 126.8±8.9 | **116.3**±8.4 | **120.2**±9.0 | **117.6**±7.1 |
| PAIRFLOW + ReDi | **84.2**±11.3 | **92.0**±6.9 | 101.1±9.6 | 98.6±8.4 | 98.8±13.9 | 95.8±8.3 | 98.9±7.5 |

Table 9: Validity scores (↑) on ZINC-250k (Irwin et al., 2012) for various methods across different steps. Best values per column are highlighted in bold.

| Method | 1 | 2 | 4 | 8 | 16 | 32 | 64 |
|---|---|---|---|---|---|---|---|
| MDLM | **15.0**±4.0 | 79.4±4.7 | 194.6±15.1 | 351.3±20.6 | 463.7±17.7 | 553.9±10.7 | 610.1±17.6 |
| UDLM | 0.3±0.5 | 65.2±8.2 | 435.7±14.4 | 775.1±19.5 | **887.3**±12.7 | **921.5**±8.5 | **937.3**±3.9 |
| Random | 0.6±0.9 | 68.3±10.7 | 351.2±15.8 | 569.4±16.6 | 611.0±16.3 | 602.4±13.3 | 571.0±13.2 |
| PAIRFLOW | 9.9±2.3 | **146.3**±10.4 | **533.9**±13.9 | **799.4**±9.2 | 873.2±14.1 | 901.0±14.2 | 907.8±7.7 |
| UDLM + DCD | 25.7±4.7 | 323.9±12.5 | 718.2±13.5 | **873.5**±8.4 | **919.8**±10.0 | **933.1**±5.9 | **942.2**±4.3 |
| PAIRFLOW + DCD | **114.9**±14.3 | **436.3**±16.5 | **725.1**±11.5 | 858.2±10.0 | 896.5±8.2 | 900.9±9.5 | 907.1±13.7 |
| UDLM + ReDi | 0.7±0.8 | 75.9±7.9 | 424.8±16.6 | 734.4±8.6 | 856.3±11.3 | 892.2±10.4 | 900.1±10.8 |
| PAIRFLOW + ReDi | **46.3**±6.3 | **221.5**±11.0 | **562.8**±12.7 | **793.6**±8.4 | **869.3**±14.2 | **897.4**±9.1 | **907.0**±10.5 |

Table 10: Uniqueness scores (↑) on ZINC-250k (Irwin et al., 2012) for various methods across different steps. Best values per column are highlighted in bold.

| Method | 1 | 2 | 4 | 8 | 16 | 32 | 64 |
|---|---|---|---|---|---|---|---|
| MDLM | 7.4±2.7 | 32.7±2.4 | 105.1±8.3 | 256.0±18.3 | 392.6±16.6 | 511.3±14.0 | 582.4±17.3 |
| UDLM | 0.3±0.5 | 65.2±8.2 | 435.7±14.4 | 775.1±19.5 | **887.3**±12.7 | **921.5**±8.5 | **937.2**±3.8 |
| Random | 0.6±0.9 | 68.3±10.7 | 351.2±15.8 | 569.4±16.6 | 611.0±16.3 | 602.4±13.3 | 571.0±13.2 |
| PAIRFLOW | **9.9**±2.3 | **146.3**±10.4 | **533.9**±13.9 | **799.4**±9.2 | 873.2±14.1 | 901.0±14.2 | 907.8±7.7 |
| UDLM + DCD | 25.7±4.7 | 323.9±12.5 | 718.2±13.5 | **873.5**±8.4 | **919.8**±10.0 | **933.1**±5.9 | **942.2**±4.3 |
| PAIRFLOW + DCD | **114.9**±14.3 | **436.3**±16.5 | **725.1**±11.5 | 858.2±10.0 | 896.5±8.2 | 900.9±9.5 | 907.1±13.7 |
| UDLM + ReDi | 0.7±0.8 | 75.9±7.9 | 424.8±16.6 | 734.3±8.7 | 856.3±11.3 | 892.2±10.4 | 900.0±10.9 |
| PAIRFLOW + ReDi | **46.3**±6.3 | **221.5**±11.0 | **562.8**±12.7 | **793.6**±8.4 | **869.3**±14.2 | **897.4**±9.1 | **907.0**±10.5 |

Table 11: Novelty scores (↑) on ZINC-250k (Irwin et al., 2012) for various methods across different steps. Best values per column are highlighted in bold.

| Method | 1 | 2 | 4 | 8 | 16 | 32 | 64 |
|---|---|---|---|---|---|---|---|
| MDLM | 3.8±2.7 | 24.2±2.6 | 84.5±7.4 | 228.3±17.1 | 372.0±15.2 | 494.5±15.2 | 569.3±17.1 |
| UDLM | 0.3±0.5 | 65.2±8.2 | 435.7±14.4 | 775.1±19.5 | **887.3**±12.7 | **921.3**±8.8 | **936.9**±4.1 |
| Random | 0.6±0.9 | 68.3±10.7 | 351.2±15.8 | 569.4±16.6 | 611.0±16.3 | 602.4±13.3 | 571.0±13.2 |
| PAIRFLOW | **9.9**±2.3 | **146.3**±10.4 | **533.9**±13.9 | **799.4**±9.2 | 873.2±14.1 | 901.0±14.2 | 907.8±7.7 |
| UDLM + DCD | 25.7±4.7 | 323.9±12.5 | 718.2±13.5 | **873.5**±8.4 | **919.7**±9.9 | **933.0**±5.8 | **942.2**±4.3 |
| PAIRFLOW + DCD | **114.9**±14.3 | **436.3**±16.5 | **725.1**±11.5 | 858.2±10.0 | 896.4±8.3 | 900.9±9.5 | 907.1±13.7 |
| UDLM + ReDi | 0.7±0.8 | 75.9±7.9 | 424.8±16.6 | 734.3±8.7 | 856.3±11.3 | 892.1±10.5 | 900.0±10.9 |
| PAIRFLOW + ReDi | **46.3**±6.3 | **221.5**±11.0 | **562.8**±12.7 | **793.5**±8.3 | **869.3**±14.2 | **897.4**±9.1 | **907.0**±10.5 |

Table 12: FID (↓) on MNIST-Binary (LeCun et al., 2002) for various methods across different steps. Best values per column are bolded.

| Method | 1 | 2 | 4 | 8 | 16 | 32 | 64 |
|---|---|---|---|---|---|---|---|
| MDLM | 204.64 | 159.26 | 103.74 | 54.41 | 28.51 | 12.31 | 7.01 |
| UDLM | 130.57 | 42.54 | 11.25 | 5.70 | **4.69** | **4.77** | **5.01** |
| Random | 128.57 | 36.59 | 9.41 | **5.60** | 5.00 | 5.10 | 5.19 |
| PAIRFLOW | **40.58** | **15.61** | **8.50** | 5.97 | 5.55 | 5.24 | 5.17 |
| UDLM + DCD | 53.84 | 16.09 | **8.06** | **7.46** | **7.12** | **6.52** | **6.65** |
| PAIRFLOW + DCD | **19.51** | **14.20** | 11.47 | 9.28 | 7.75 | 7.82 | 8.42 |
| UDLM + ReDi | 18.44 | 10.35 | **8.11** | **6.65** | **6.56** | **6.55** | **6.49** |
| PAIRFLOW + ReDi | **12.90** | **9.38** | 8.54 | 6.85 | 6.79 | 6.96 | 6.94 |

# E  EXPERIMENT ON CONTINUOUS FLOW MATCHING

Alongside our main experiments in the discrete setting, we also demonstrate the potential of our method to extend to continuous domains, as illustrated by the toy experiment presented below. Here, we denote by PAIRFLOW a continuous flow model trained on source–target pairs constructed using the continuous variant of the algorithm described in Sec. 4.2.

## E.1  CLOSED-FORM VELOCITY IN CONTINUOUS FLOW MATCHING

**Setup.** Let $X_0 \sim p_0$ (source), $X_1 \sim q$ (target) be independent random variables in $\mathbb{R}^N$ and consider the linear probability path

$$X_t = (1 - t)X_0 + tX_1, \qquad t \in [0, 1]. \tag{61}$$

Table 13: FID ($\downarrow$) and IS ($\uparrow$) on CIFAR-10 (Krizhevsky et al., 2009) for various methods across different timesteps. Best values per column are bolded.

| | 1 | 2 | 4 | 8 | 16 | 32 | 64 | 128 | 256 | 512 | 1024 |
|---|---|---|---|---|---|---|---|---|---|---|---|
| Method | | | | | | FID ($\downarrow$) | | | | | |
| MDLM | 407.31 | 359.97 | 340.43 | 340.98 | 321.79 | 228.34 | 131.63 | 77.44 | 52.89 | 42.88 | 39.04 |
| UDLM | 340.47 | 321.98 | 255.95 | 151.60 | 91.31 | 62.43 | 49.06 | 42.26 | 40.18 | 37.63 | 37.60 |
| Random | 328.83 | 314.64 | 220.16 | 127.66 | 82.76 | 58.42 | 45.18 | 39.50 | 36.78 | 35.16 | 35.07 |
| PAIRFLOW | **269.87** | **260.29** | **192.62** | **112.78** | **73.87** | **52.22** | **40.06** | **34.42** | **33.83** | **32.42** | **31.85** |
| UDLM + DCD | 318.33 | 282.38 | 204.15 | 108.30 | **75.56** | 69.87 | 74.62 | 79.35 | 84.34 | 85.79 | 87.87 |
| PAIRFLOW + DCD | **223.86** | **190.15** | **139.46** | **95.91** | 82.11 | **80.70** | **86.59** | **100.16** | **114.25** | **123.41** | **129.45** |
| UDLM + ReDi | **251.02** | **212.20** | **171.24** | 148.18 | 138.18 | 131.94 | 125.81 | 123.77 | 123.39 | 121.52 | 121.06 |
| PAIRFLOW + ReDi | 275.09 | 250.87 | 184.45 | **119.04** | **89.46** | **74.46** | **66.22** | **61.83** | **61.35** | **60.10** | **59.53** |
| Method | | | | | | IS ($\uparrow$) | | | | | |
| MDLM | 1.21 | 1.22 | 1.24 | 1.31 | 1.52 | 2.47 | 4.03 | 5.08 | 5.66 | 6.18 | 6.38 |
| UDLM | 1.32 | 1.48 | 2.11 | 3.54 | 4.85 | 5.90 | 6.25 | 6.56 | 6.66 | 6.87 | 6.81 |
| Random | 1.37 | 1.52 | 2.47 | 4.01 | 5.24 | 6.08 | 6.56 | 6.75 | 6.87 | 7.07 | 7.05 |
| PAIRFLOW | **1.72** | **1.80** | **2.70** | **4.27** | **5.61** | **6.23** | **6.86** | **7.00** | **7.12** | **7.14** | **7.33** |
| UDLM + DCD | 1.49 | 1.60 | 2.23 | 3.90 | **5.09** | **5.45** | **5.42** | **5.16** | **5.09** | **5.07** | **4.84** |
| PAIRFLOW + DCD | **2.21** | **2.38** | **3.23** | **4.38** | 4.99 | 5.13 | 4.97 | 4.48 | 4.26 | 3.82 | 3.81 |
| UDLM + ReDi | **1.85** | **2.35** | **2.96** | 3.37 | 3.52 | 3.67 | 3.86 | 3.96 | 3.93 | 3.99 | 3.94 |
| PAIRFLOW + ReDi | 1.80 | 1.95 | 2.91 | **4.18** | **5.02** | **5.65** | **5.88** | **6.03** | **6.06** | **6.09** | **6.30** |

Table 14: FID of PAIRFLOW on MNIST (LeCun et al., 2002) with continuous values, measured using FID over 50K samples across various timesteps. Best values are bolded.

| Method | 1 | 2 | 4 | 8 | 16 | 32 | 64 |
|---|---|---|---|---|---|---|---|
| CondOT | 398.43 | 91.17 | 27.34 | 10.80 | 5.81 | 3.99 | 3.16 |
| PAIRFLOW | **74.89** | **14.40** | **6.89** | **3.78** | **2.70** | **2.42** | **2.37** |
| CondOT+RF | 32.70 | 9.12 | 5.46 | 4.24 | 3.93 | 3.56 | 3.24 |
| PAIRFLOW+RF | **28.61** | **4.15** | **3.01** | **2.87** | **2.89** | **2.91** | **2.94** |

For flow matching with the linear path Eqn. 61, the optimal velocity field equals the conditional drift:

$$v_t(x) = \mathbb{E}\left[X_1 - X_0 | X_t = x\right]. \tag{62}$$

We derive a closed form of Eqn. 62 that is directly computable from $p_0$ and $q$.

**Derivation.** By Bayes' rule with a Dirac constraint for the linear relation Eqn. 61,

$$p(x_0, x_1 \mid X_t = x) \quad \propto \quad p_0(x_0)\, q(x_1)\, \delta\left(x - (1-t)x_0 - tx_1\right). \tag{63}$$

Integrating out $x_0$ using $\delta(Ay - b) = |\det A|^{-1}\, \delta\left(y - A^{-1}b\right)$ with $A = (1-t)I$ gives

$$p(x_1 \mid X_t = x) \quad \propto \quad q(x_1)\, (1-t)^{-D}\, p_0\left(\frac{x - tx_1}{1-t}\right). \tag{64}$$

Hence,

$$v_t(x) = \frac{\iint (x_1 - x_0)\, p(x_0, x_1 \mid X_t = x)\, \mathrm{d}x_0 \mathrm{d}x_1}{\iint p(x_0, x_1 \mid X_t = x)\, \mathrm{d}x_0\, \mathrm{d}x_1} \tag{65}$$

$$= \frac{\iint (x_1 - x_0)\, p_0(x_0)\, q(x_1)\, \delta\left(x - (1-t)x_0 - tx_1\right) \mathrm{d}x_0 \mathrm{d}x_1}{\iint p_0(x_0)\, q(x_1)\, \delta\left(x - (1-t)x_0 - tx_1\right)\, \mathrm{d}x_0\, \mathrm{d}x_1} \tag{66}$$

$$= \frac{\int q(x_1)\, p_0\left(\frac{x - tx_1}{1-t}\right)\left(x_1 - \frac{x - tx_1}{1-t}\right) \mathrm{d}x_1}{\int q(x_1)\, p_0\left(\frac{x - tx_1}{1-t}\right), \mathrm{d}x_1}. \tag{67}$$

Observing $x_1 - \frac{x - tx_1}{1-t} = \frac{x_1 - x}{1-t}$, we obtain the compact form

$$v_t(x) = \frac{1}{1-t} \frac{\int q(x_1)\, p_0\left(\frac{x - tx_1}{1-t}\right)(x_1 - x)\, \mathrm{d}x_1}{\int q(x_1)\, p_0\left(\frac{x - tx_1}{1-t}\right)\, \mathrm{d}x_1}. \tag{68}$$

When we have a dataset with samples $\{d_m\}_{m=1}^M$, the target distribution $q$ is approximated by the empirical measure $q(x_1) \approx \frac{1}{M} \sum_{m=1}^M \delta_{d_m}(x_1)$, then Eqn. 68 reduces as follow:

$$v_t(x) = \frac{1}{1-t} \frac{\sum_{m=1}^M p_0\left(\frac{x - td_m}{1-t}\right)(d_m - x)}{\sum_{m=1}^M p_0\left(\frac{x - td_m}{1-t}\right)}. \tag{69}$$

When $p_0$ is standard Gaussian, $p_0(y) = (2\pi)^{-D/2} \exp\left(-\frac{1}{2}\|y\|_2^2\right)$, the normalizing constants cancel in Eqn. 69, yielding the closed form velocity:

$$v_t(x) = \frac{1}{1-t} \frac{\sum_{m=1}^M \exp\left(-\frac{1}{2}\left\|\frac{x - td_m}{1-t}\right\|_2^2\right)(d_m - x)}{\sum_{m=1}^M \exp\left(-\frac{1}{2}\left\|\frac{x - td_m}{1-t}\right\|_2^2\right)}. \tag{70}$$

This formulation has already been introduced in previous works (Karras et al., 2022; Bertrand et al., 2025); however, to the best of our knowledge, no prior work has extended this idea to designing couplings for accelerating flow models using the re-flow technique (Liu & Gong, 2023). In the continuous domain, the backward velocity can be obtained directly by flipping the sign of the forward velocity. In contrast, in the discrete domain, the corresponding expression does not converge as $\lim_{t \to 1}$, and thus the backward velocity cannot be employed for sampling starting from data points. Therefore, in this section, we perform experiments using the forward velocity.

### E.2 CONTINUOUS FLOW MATCHING ON MNIST

We train rectified flow models on MNIST (LeCun et al., 2002) using two pairing strategies: (i) independent pairing (baseline) and (ii) closed-form pairing as described in Sec. E.1. We adopt CondOT (Lipman et al., 2023) as our base flow model, which is originally trained with a independent pairing. We denote the variant of CondOT trained on pairs generated by the closed-form forward velocity as PAIRFLOW. To enable a few-step sampling, we subsequently apply rectification distillation (ReFlow (Liu & Gong, 2023)) to each pretrained model, denoted by the suffix "+RF".

We use an NCSN++-style U-Net backbone (Song et al., 2021) with a base width of 64 and 3 downsampling stages (doubling channels at each stage), optimized using Adam (Kingma & Ba, 2014)

with a learning rate of $2 \times 10^{-4}$. The pretraining takes 500 epochs. The distillation stage requires 200 epochs with a learning rate of $2 \times 10^{-5}$.

Tab. 14 summarizes performance at various sampling steps. Without distillation, closed-form pairing (PAIRFLOW) yields significantly better FID in the few-step settings and maintains the performance in the many-step settings, relative to the baseline. With distillation (ReFlow (Liu & Gong, 2023), our method still shows better performance: PAIRFLOW+RF achieves a lower FID in every sampling budget than ReFlow applied to the baseline. These results show that closed-form pairing benefits both undistilled and distilled flow models, with especially large gains when the sampling steps are small.

### E.3 CONTINUOUS RECTIFIED FLOWS ON DIMENSION-VARYING SYNTHETIC DATA

To assess scalability, we construct an N-fold product of the standard two-moons distribution, yielding a dataset in $\mathbb{R}^{2N}$. We consider dimensions $d \in \{2, 4, 8, 16, 32, 64, 128, 256\}$ (i.e., $d = 2N$) and train rectified flow models with and without closed-form pairing under a common training setup. The architecture is a simple transformer-based encoder with depth 8, where the hidden size increases with dimension as $32, 64, 128, 192, 256, 384, 512, 768$, respectively.

For the synthetic experiments we report the Chamfer distance (log scale) between 50,000 training datapoints and 5,000 generated samples. Since the dataset is an $N$-fold product of 2D two-moons, Chamfer distance is computed using only the first two coordinates to keep the metric scale consistent across $d$ and measure fidelity to the base 2D geometry.

Fig. 7 shows the quantitative results. At low dimensions, closed-form pairing yields substantial improvements over the independently paired baseline. However, as the data dimension increases, we observe that the magnitude of the improvement decreases. This trend suggests a practical limitation of closed-form pairing for high-dimensional continuous data.

## F    ADDITIONAL QUALITATIVE RESULTS

In addition to Fig. 4 in the main paper, we further visualize the 1-step and 2-step generation results for MNIST-Binary (LeCun et al., 2002) in Fig. 9 and Fig. 10. As discussed in Sec. 5.3, PAIRFLOW outperforms the base models (Schiff et al., 2025; Sahoo et al., 2024) and achieves comparable quality to the base models combined with acceleration methods (Sahoo et al., 2025; Yoo et al., 2025). Additional visualizations for CIFAR-10 (Krizhevsky et al., 2009) with 64- and 256-step generations are shown in Fig. 8, demonstrating that our method outperforms the other base models.

## G    REFLOW ITERATION RESULTS

This section details the results of applying the iterative reflow procedure (Liu & Gong, 2023; Yoo et al., 2025) on the QM9 dataset (Ramakrishnan et al., 2014). We generated 1,024 samples across various timesteps and reflow iterations, assessing validity, uniqueness, and novelty. The configuration for subsequent reflow iterations follows the same protocol as our main experiment. As shown in Tab. 15, Tab. 16, and Tab. 17, all metrics are averaged over 10 independent runs with standard deviations provided. Additional reflow steps are denoted by "+ ReDi" (Yoo et al., 2025). We observed that iterative reflow consistently enhances few-step generation capabilities. Notably, PAIRFLOW demonstrates superior performance over the baseline; it outperforms UDLM not only at equivalent iteration levels but also surpasses UDLM with multiple reflow iterations even when PAIRFLOW uses one or no additional iterations.

Table 15: validity scores (↑) on QM9 (Ramakrishnan et al., 2014) for UDLM and PAIRFLOW across varying rectification steps and NFEs (1 to 64). The best and second-best values per column are highlighted in **bold** and underlined, respectively.

| Method | 1 | 2 | 4 | 8 | 16 | 32 | 64 |
|---|---|---|---|---|---|---|---|
| UDLM | 17.5±3.2 | 125.5±11.8 | 497.6±8.3 | 826.6±10.3 | 953.5±6.1 | **991.9±4.2** | 1000.1±3.5 |
| + ReDi | 59.7±8.8 | 232.4±9.2 | 588.4±15.8 | 849.6±14.2 | 940.5±8.5 | 967.5±5.2 | 978.8±5.2 |
| + ReDi | 160.9±11.8 | 368.0±11.8 | 673.3±11.2 | 878.5±11.6 | 945.4±8.1 | 967.9±7.4 | 975.8±8.0 |
| + ReDi | 280.2±6.7 | 470.7±18.6 | 742.2±18.7 | 897.9±9.4 | 945.3±5.2 | 965.0±5.9 | 972.6±7.1 |
| PAIRFLOW | 223.4±12.7 | 416.0±12.4 | 734.9±7.2 | 921.5±11.0 | **977.1±3.9** | 990.9±5.9 | **1000.2±4.5** |
| + ReDi | 361.0±115.2 | 512.6±44.2 | 775.7±10.0 | 929.1±11.6 | 976.2±4.5 | 985.6±6.7 | 993.1±7.1 |
| + ReDi | 443.4±13.4 | 598.6±18.4 | 823.0±16.0 | 935.2±7.7 | 969.0±5.7 | 984.1±6.0 | 989.0±3.8 |
| + ReDi | **529.2±10.6** | **688.0±11.3** | **863.6±9.5** | **945.9±4.5** | 975.7±6.1 | 982.6±7.1 | 990.2±7.2 |

Table 16: Uniqueness scores (↑) on QM9 (Ramakrishnan et al., 2014) for UDLM and PAIRFLOW across varying rectification steps and NFEs (1 to 64). The best and second-best values per column are highlighted in **bold** and underlined, respectively.

| Method | 1 | 2 | 4 | 8 | 16 | 32 | 64 |
|---|---|---|---|---|---|---|---|
| UDLM | 17.5±3.2 | 125.4±11.7 | 495.0±8.2 | 819.5±11.4 | 943.0±5.7 | 979.1±5.0 | 990.0±4.7 |
| + ReDi | 59.7±8.8 | 231.6±9.5 | 581.0±15.1 | 834.7±11.4 | 917.5±9.6 | 944.8±6.2 | 956.1±5.2 |
| + ReDi | 159.4±10.9 | 363.3±12.2 | 657.2±9.3 | 846.3±11.8 | 910.8±7.3 | 930.5±9.3 | 938.8±11.7 |
| + ReDi | 275.6±7.2 | 456.3±17.7 | 712.1±15.7 | 856.8±10.3 | 899.0±4.9 | 907.0±10.9 | 918.9±5.7 |
| PAIRFLOW | 223.0±12.3 | 414.7±12.0 | 731.4±6.9 | **917.4±11.8** | **971.6±4.3** | **986.2±5.3** | **994.8±5.2** |
| + ReDi | 359.5±113.1 | 507.7±43.0 | 765.3±8.8 | 913.5±10.2 | 959.7±5.8 | 968.8±8.4 | 973.0±9.6 |
| + ReDi | 437.9±13.8 | 586.7±17.9 | 801.0±15.4 | 906.5±10.1 | 939.1±6.2 | 955.9±7.2 | 960.2±6.5 |
| + ReDi | **516.9±10.8** | **662.8±13.4** | **828.7±8.4** | 903.2±4.7 | 932.5±9.5 | 942.8±4.8 | 949.6±8.4 |

Table 17: Novelty scores (↑) on QM9 (Ramakrishnan et al., 2014) for UDLM and PAIRFLOW across varying rectification steps and NFEs (1 to 64). The best and second-best values per column are highlighted in **bold** and underlined, respectively.

| Method | 1 | 2 | 4 | 8 | 16 | 32 | 64 |
|---|---|---|---|---|---|---|---|
| UDLM | 13.8±2.9 | 52.0±8.2 | 120.0±3.8 | **152.4±9.1** | **144.2±12.4** | **147.2±9.7** | **145.1±9.0** |
| + ReDi | 31.4±8.0 | 73.3±7.1 | 110.4±12.2 | 126.8±8.9 | 116.3±8.4 | 120.2±9.0 | 117.6±7.1 |
| + ReDi | 61.1±6.5 | 103.0±8.6 | **129.0±9.7** | 124.6±12.5 | 128.3±5.6 | 128.4±10.6 | 122.8±9.4 |
| + ReDi | 91.2±9.2 | **116.4±8.1** | 127.2±8.3 | 124.2±9.3 | 121.9±6.0 | 117.0±4.9 | 121.8±8.0 |
| PAIRFLOW | 68.8±7.8 | 85.6±10.0 | 109.2±7.8 | 96.8±9.9 | 106.5±12.5 | 108.9±9.4 | 110.0±9.9 |
| + ReDi | 84.2±11.3 | 92.0±6.9 | 101.1±9.6 | 98.6±8.4 | 98.8±13.9 | 95.8±8.3 | 98.9±7.5 |
| + ReDi | 100.8±7.7 | 109.5±8.7 | 101.0±6.8 | 101.1±7.8 | 94.5±6.6 | 95.6±9.6 | 99.0±8.8 |
| + ReDi | **114.6±9.4** | 108.0±5.9 | 106.5±8.9 | 96.2±9.8 | 95.6±9.0 | 94.8±7.1 | 95.7±8.2 |

## H  SUBSET PAIRING RESULTS

In this section, we present comprehensive experimental results from applying our subset-partition pairing technique to the ZINC-250k molecular dataset (Irwin et al., 2012). Following the same protocol as our main experiments, we generated 1,024 samples across varying timesteps and partition counts to evaluate validity, uniqueness, and novelty. The only deviation from the standard method is the pairing strategy; here, we calculate the closed-form backward velocity exclusively within each subset to reduce the computational cost of Eqn. 10. All reported metrics, summarized in Tab. 18, Tab. 19, and Tab. 20, are averaged over 10 independent runs with corresponding standard deviations. Additionally, we report the pairing-time cost for each subset-partition configuration. The results demonstrate that our subset-pairing algorithm effectively reduces the computational time for pairing, while maintaining performance comparable to the full-set baseline.

Table 18: Validity scores ($\uparrow$) on the Zinc-250k dataset (Irwin et al., 2012) evaluated across different subset partitions and NFEs (1 to 64). The best and second-best values per column are highlighted in **bold** and underlined, respectively. $T_{\text{PAIRFLOW}}$ denotes the runtime (in minutes) of each configuration, measured in wall-clock time using an RTX A6000 GPU.

| Method | $T_{\text{PAIRFLOW}}$ | 1 | 2 | 4 | 8 | 16 | 32 | 64 |
|---|---|---|---|---|---|---|---|---|
| UDLM | 0 | 0.3±0.5 | 65.2±8.2 | 435.7±14.4 | 775.1±19.5 | **887.3±12.7** | **921.5±8.5** | **937.3±3.9** |
| Random | 0 | 0.6±0.9 | 68.3±10.7 | 351.2±15.8 | 569.4±16.6 | 611.0±16.3 | 602.4±13.3 | 571.0±13.2 |
| PAIRFLOW (Full) | 13m | 9.9±2.3 | **146.3±10.4** | **533.9±13.9** | 799.4±9.2 | 873.2±14.1 | 901.0±14.2 | 907.8±7.7 |
| PAIRFLOW (2-Sub) | 6m | 10.6±2.8 | 145.7±13.5 | 530.6±20.3 | **802.5±6.9** | 882.6±7.1 | 902.4±13.4 | 911.5±9.2 |
| PAIRFLOW (4-Sub) | 2.9m | 12.1±3.3 | 142.5±5.2 | 509.7±11.0 | 780.9±14.4 | 858.9±7.9 | 886.7±8.7 | 899.0±10.5 |
| PAIRFLOW (8-Sub) | 1.5m | **12.3±2.5** | 141.1±6.7 | 510.9±17.2 | 766.8±12.6 | 857.5±8.4 | 886.9±9.6 | 896.5±6.3 |

Table 19: Uniqueness scores ($\uparrow$) on Zinc-250k (Irwin et al., 2012) evaluated across different subset partitions and NFEs (1 to 64). The best and second-best values per column are highlighted in **bold** and underlined, respectively. $T_{\text{PAIRFLOW}}$ denotes the runtime (in minutes) of each configuration.

| Method | $T_{\text{PAIRFLOW}}$ | 1 | 2 | 4 | 8 | 16 | 32 | 64 |
|---|---|---|---|---|---|---|---|---|
| UDLM | 0 | 0.3±0.5 | 65.2±8.2 | 435.7±14.4 | 775.1±19.5 | **887.3±12.7** | **921.5±8.5** | **937.2±3.8** |
| Random | 0 | 0.6±0.9 | 68.3±10.7 | 351.2±15.8 | 569.4±16.6 | 611.0±16.3 | 602.4±13.3 | 571.0±13.2 |
| PAIRFLOW (Full) | 13m | 9.9±2.3 | **146.3±10.4** | **533.9±13.9** | 799.4±9.2 | 873.2±14.1 | 901.0±14.2 | 907.8±7.7 |
| PAIRFLOW (2-Sub) | 6m | 10.6±2.8 | 145.7±13.5 | 530.6±20.3 | **802.5±6.9** | 882.6±7.1 | 902.4±13.4 | 911.5±9.2 |
| PAIRFLOW (4-Sub) | 2.9m | 12.1±3.3 | 142.5±5.2 | 509.7±11.0 | 780.9±14.4 | 858.9±7.9 | 886.7±8.7 | 899.0±10.5 |
| PAIRFLOW (8-Sub) | 1.5m | **12.3±2.5** | 141.1±6.7 | 510.9±17.2 | 766.8±12.6 | 857.5±8.4 | 886.9±9.6 | 896.5±6.3 |

Table 20: Novelty scores ($\uparrow$) on Zinc-250k (Irwin et al., 2012) evaluated across different subset partitions and NFEs (1 to 64). The best and second-best values per column are highlighted in **bold** and underlined, respectively. $T_{\text{PAIRFLOW}}$ denotes the runtime (in minutes) of each configuration.

| Method | $T_{\text{PAIRFLOW}}$ | 1 | 2 | 4 | 8 | 16 | 32 | 64 |
|---|---|---|---|---|---|---|---|---|
| UDLM | 0 | 0.3±0.5 | 65.2±8.2 | 435.7±14.4 | 775.1±19.5 | **887.3±12.7** | **921.3±8.8** | **936.9±4.1** |
| Random | 0 | 0.6±0.9 | 68.3±10.7 | 351.2±15.8 | 569.4±16.6 | 611.0±16.3 | 602.4±13.3 | 571.0±13.2 |
| PAIRFLOW (Full) | 13m | 9.9±2.3 | **146.3±10.4** | **533.9±13.9** | 799.4±9.2 | 873.2±14.1 | 901.0±14.2 | 907.8±7.7 |
| PAIRFLOW (2-Sub) | 6m | 10.6±2.8 | 145.7±13.5 | 530.6±20.3 | **802.5±6.9** | 882.6±7.1 | 902.4±13.4 | 911.5±9.2 |
| PAIRFLOW (4-Sub) | 2.9m | 12.1±3.3 | 142.5±5.2 | 509.7±11.0 | 780.9±14.4 | 858.8±8.0 | 886.7±8.7 | 899.0±10.5 |
| PAIRFLOW (8-Sub) | 1.5m | **12.3±2.5** | 141.1±6.7 | 510.9±17.2 | 766.8±12.6 | 857.5±8.4 | 886.9±9.6 | 896.5±6.3 |

## I APPLICATION FOR MORE COMPLEX SYSTEMS

In this section, we evaluate our method on a higher-dimensional dataset. Specifically, we use the FFHQ (Karras et al., 2019) dataset, downsampled to $64 \times 64$. Following the same protocol as in our main experiments, we generate 5,000 samples across varying timesteps and report the FID computed against the training set. The results of this experiment are provided in Tab. 21. All training hyperparameters are kept identical to those used in the CIFAR-10 experiments described in Section 5.

Table 21: Comparison of FID scores ($\downarrow$) on FFHQ (Karras et al., 2019) downsampled to $64 \times 64$ resolution across extended NFE steps (1 to 1024). Best values per column are highlighted in bold.

| Method | 1 | 2 | 4 | 8 | 16 | 32 | 64 | 128 | 256 | 512 | 1024 |
|---|---|---|---|---|---|---|---|---|---|---|---|
| UDLM | 403.04 | 399.26 | 363.97 | 273.31 | 153.71 | 97.87 | 74.85 | 63.93 | 59.28 | 55.99 | 55.30 |
| PAIRFLOW | **394.14** | **368.36** | **329.13** | **243.88** | **140.05** | **90.85** | **69.67** | **59.86** | **56.52** | **54.19** | **53.18** |

We adopt the LM1B (Chelba et al., 2014) dataset to evaluate our method under a substantially larger vocabulary size and training corpus. The text corpus is segmented into sequences of varying lengths ($N = 16, 32, 64, 128$), while keeping the total number of training samples fixed ($|X_1| \approx 3.5M$). To assess generation quality, we compute generative perplexity using GPT-2 Large and entropy on 1,024 generated samples for each NFE setting. The results are summarized in Tab. 22 and Tab. 23. For training, we follow the network hyperparameter configuration of (Schiff et al., 2025), modifying only the number of training iterations for each sequence dimensionality.

Table 22: Generative Perplexity ($\downarrow$) on LM1B (Chelba et al., 2014) measured with GPT2-large across varying lengths ($N$) and their corresponding training iterations (Iter.) over NFE steps 4 to 1024. Best values are highlighted in bold.

| $N$ | Iter. | Method | 4 | 8 | 16 | 32 | 64 | 128 | 256 | 512 | 1024 |
|---|---|---|---|---|---|---|---|---|---|---|---|
| 16 | 200k | UDLM | 299.18 | 225.92 | 207.17 | 195.82 | 200.77 | **197.04** | 199.12 | **195.37** | 198.22 |
| | | PAIRFLOW | **242.22** | **208.04** | **200.99** | **190.36** | **191.74** | 199.45 | **188.84** | 196.91 | **198.12** |
| 32 | 200k | UDLM | 263.93 | 192.78 | 167.85 | 167.49 | 155.68 | 150.52 | 152.40 | 151.74 | 154.02 |
| | | PAIRFLOW | **218.48** | **172.27** | **156.35** | **150.53** | **143.83** | **145.77** | **142.54** | **141.04** | **147.57** |
| 64 | 400k | UDLM | 214.07 | 150.59 | 130.49 | 120.19 | 117.90 | 116.23 | 112.24 | 113.77 | 115.11 |
| | | PAIRFLOW | **174.78** | **138.94** | **123.06** | **115.71** | **114.73** | **112.92** | **111.29** | **107.06** | **110.83** |
| 128 | 600k | UDLM | 169.61 | 123.48 | 105.13 | 98.94 | 97.89 | 94.92 | 93.75 | 94.12 | 93.59 |
| | | PAIRFLOW | **167.90** | **121.09** | **102.16** | **96.61** | **93.93** | **91.51** | **90.21** | **89.09** | **89.07** |

Table 23: Comparison of Entropy ($\uparrow$) on LM1B (Chelba et al., 2014) across varying lengths ($N$) and training iterations (Iter.) over NFE steps 4 to 1024. Best values are highlighted in bold.

| $N$ | Iter. | Method | 4 | 8 | 16 | 32 | 64 | 128 | 256 | 512 | 1024 |
|---|---|---|---|---|---|---|---|---|---|---|---|
| 16 | 200k | UDLM | 2.46 | **2.49** | 2.50 | 2.50 | 2.50 | 2.51 | 2.50 | 2.51 | 2.50 |
| | | PAIRFLOW | **2.48** | **2.49** | **2.51** | **2.52** | **2.52** | **2.53** | **2.52** | **2.53** | **2.52** |
| 32 | 200k | UDLM | 3.05 | 3.09 | 3.12 | 3.13 | 3.13 | 3.13 | 3.13 | 3.13 | 3.13 |
| | | PAIRFLOW | **3.06** | **3.12** | **3.13** | **3.14** | **3.15** | **3.15** | **3.15** | **3.15** | **3.16** |
| 64 | 400k | UDLM | **3.57** | 3.63 | 3.67 | 3.68 | 3.69 | 3.70 | 3.70 | 3.69 | 3.70 |
| | | PAIRFLOW | **3.57** | **3.65** | **3.69** | **3.70** | **3.71** | **3.71** | **3.71** | **3.72** | **3.71** |
| 128 | 600k | UDLM | 3.98 | 4.09 | 4.14 | 4.16 | 4.17 | 4.17 | 4.17 | 4.18 | 4.18 |
| | | PAIRFLOW | **4.00** | **4.11** | **4.16** | **4.18** | **4.19** | **4.20** | **4.20** | **4.19** | **4.20** |

## J  ANALYSIS FOR THE OVERFITTING IN IMAGE DOMAINS

In this section, we evaluate our method using FID computed on the test sets of two image domains: CIFAR-10 (Krizhevsky et al., 2009) and MNIST-Binary (LeCun et al., 2002). For CIFAR-10, we additionally report FID scores measured with DINOv2 (Oquab et al., 2024). The overall results are summarized in Tab. 24 and Tab. 25. Across all evaluation metrics, the performance trend is consistent with our main findings—PAIRFLOW delivers improved generation quality over the baseline, with especially strong gains in the few-step generation regime.

Table 24: FID and FID-Dino scores ($\downarrow$) on test dataset for CIFAR-10 (Krizhevsky et al., 2009) comparison across extended NFE steps (1 to 1024). Best values are bolded.

| NFE | 1 | 2 | 4 | 8 | 16 | 32 | 64 | 128 | 256 | 512 | 1024 |
|---|---|---|---|---|---|---|---|---|---|---|---|
| Method | | | | | FID ($\downarrow$) | | | | | | |
| UDLM | 306.45 | 296.77 | 266.64 | 178.11 | 114.04 | 80.40 | 62.70 | 53.83 | 50.96 | 47.61 | 47.16 |
| PAIRFLOW | **235.65** | **247.18** | **209.94** | **137.16** | **94.24** | **67.53** | **51.48** | **43.79** | **42.44** | **40.85** | **39.92** |
| Method | | | | | FID-DINOv2 ($\downarrow$) | | | | | | |
| UDLM | 2448.46 | 2410.65 | 1975.58 | 1344.44 | 959.39 | 755.80 | 646.47 | 598.53 | 598.54 | 553.17 | 560.75 |
| PAIRFLOW | **2059.26** | **1988.21** | **1626.08** | **1127.30** | **828.09** | **623.89** | **530.70** | **486.59** | **484.27** | **470.33** | **470.05** |

Table 25: FID scores (↓) on the MNIST-Binary (LeCun et al., 2002) test set across various NFE steps (1 to 64). Best values are bolded.

| Method | 1 | 2 | 4 | 8 | 16 | 32 | 64 |
|---|---|---|---|---|---|---|---|
| UDLM | 129.05 | 42.17 | 11.42 | **6.18** | **5.13** | **5.37** | 5.50 |
| PAIRFLOW | **42.87** | **17.37** | **9.62** | 6.36 | 5.80 | 5.51 | **5.24** |
| UDLM+DCD | 57.85 | 19.23 | **9.82** | **8.41** | 7.87 | **7.12** | 7.38 |
| PAIRFLOW+DCD | **19.56** | **13.06** | 10.90 | 8.55 | **7.40** | 7.22 | 7.85 |
| UDLM+ReDi | 19.08 | 10.79 | **8.77** | **7.01** | **6.89** | **6.57** | **6.61** |
| PAIRFLOW+ReDi | **13.73** | **9.59** | 8.98 | 7.24 | 7.22 | 6.98 | 7.12 |

We further assess potential memorization by measuring nearest-neighbor distances with respect to the training set. For MNIST-Binary (LeCun et al., 2002), we compute pixel-wise $\ell_2$ distances, whereas for CIFAR-10 (Krizhevsky et al., 2009), we evaluate both $\ell_2$ distance and cosine similarity between features extracted using DINOv2 (Oquab et al., 2024). As summarized in Tab. 26 and Tab. 27, across all evaluation settings, the nearest-neighbor distances of PAIRFLOW are comparable to or slightly larger than those of the baseline. These results support the conclusion that our method does not suffer from severe overfitting or excessive memorization of the training data.

Table 26: Comparison of $\ell_2$ and Dino (Oquab et al., 2024) Cosine nearest neighbor distance on the CIFAR-10 (Krizhevsky et al., 2009) training set across extended NFE steps (1 to 1024). Best values are bolded.

| NFE | 1 | 2 | 4 | 8 | 16 | 32 | 64 | 128 | 256 | 512 | 1024 |
|---|---|---|---|---|---|---|---|---|---|---|---|
| Metric | $\ell_2$ (↑) | | | | | | | | | | |
| UDLM | 7.97 | **9.13** | **10.06** | **10.06** | **9.40** | **8.94** | **8.63** | 8.41 | 8.29 | 8.29 | 8.22 |
| PAIRFLOW | **8.03** | 8.76 | 9.42 | 9.56 | 9.18 | 8.75 | **8.63** | **8.55** | **8.51** | **8.52** | **8.52** |
| Metric | Cosine(DINOv2) (↓) | | | | | | | | | | |
| UDLM | **0.242** | **0.227** | **0.231** | 0.241 | 0.237 | 0.238 | 0.235 | 0.237 | 0.237 | **0.235** | 0.237 |
| PAIRFLOW | 0.245 | 0.228 | 0.235 | **0.239** | **0.236** | **0.232** | **0.232** | **0.230** | **0.232** | 0.235 | **0.233** |

Table 27: Comparison of $\ell_2$ nearest neighbor distance on the MNIST-Binary (LeCun et al., 2002) training set across extended NFE steps (1 to 64). Best values are bolded.

| Method | 1 | 2 | 4 | 8 | 16 | 32 | 64 |
|---|---|---|---|---|---|---|---|
| UDLM | **8.18** | **7.06** | 6.55 | **6.42** | **6.27** | 6.29 | 6.25 |
| PAIRFLOW | 7.36 | 6.95 | **6.63** | **6.42** | 6.26 | **6.34** | **6.30** |
| UDLM+DCD | 7.24 | 6.78 | 6.57 | 6.50 | 6.31 | 6.40 | 6.39 |
| PAIRFLOW+DCD | **7.64** | **7.34** | **7.11** | **6.91** | **6.74** | **6.76** | **6.79** |
| UDLM+ReDi | **7.14** | **6.81** | **6.49** | **6.24** | **6.10** | **6.14** | **6.08** |
| PAIRFLOW+ReDi | 6.84 | 6.57 | 6.35 | 6.09 | 5.95 | 5.97 | 5.96 |

# K  LIMITATIONS AND FUTURE WORK

We hope this work initiates broader discussion on reducing training compute while still enabling fast generation in generative models. Such efficiency can have a significant impact, from reducing energy consumption in training large-scale generative models to contributing to the democratization of foundation model development.

A natural follow-up question to our work is whether the same idea can be applied to continuous Flow Matching (FM). We have evaluated this extension on continuous FM models, with results provided in App. E. Our experiments with synthetic data show that the method is effective for

relatively low-dimensional data, while its advantage a bit diminishes for higher-dimensional data. We will further investigate the effect of our method on continuous data, where we hypothesize that a substantially larger number of source–target pairs will be required. Nonetheless, we emphasize that even in this initial exploration of accelerating flow models through well-aligned pairing, PAIRFLOW is particularly well-suited for low-dimensional discrete data, which includes many forms of scientific data such as molecular and protein structures.

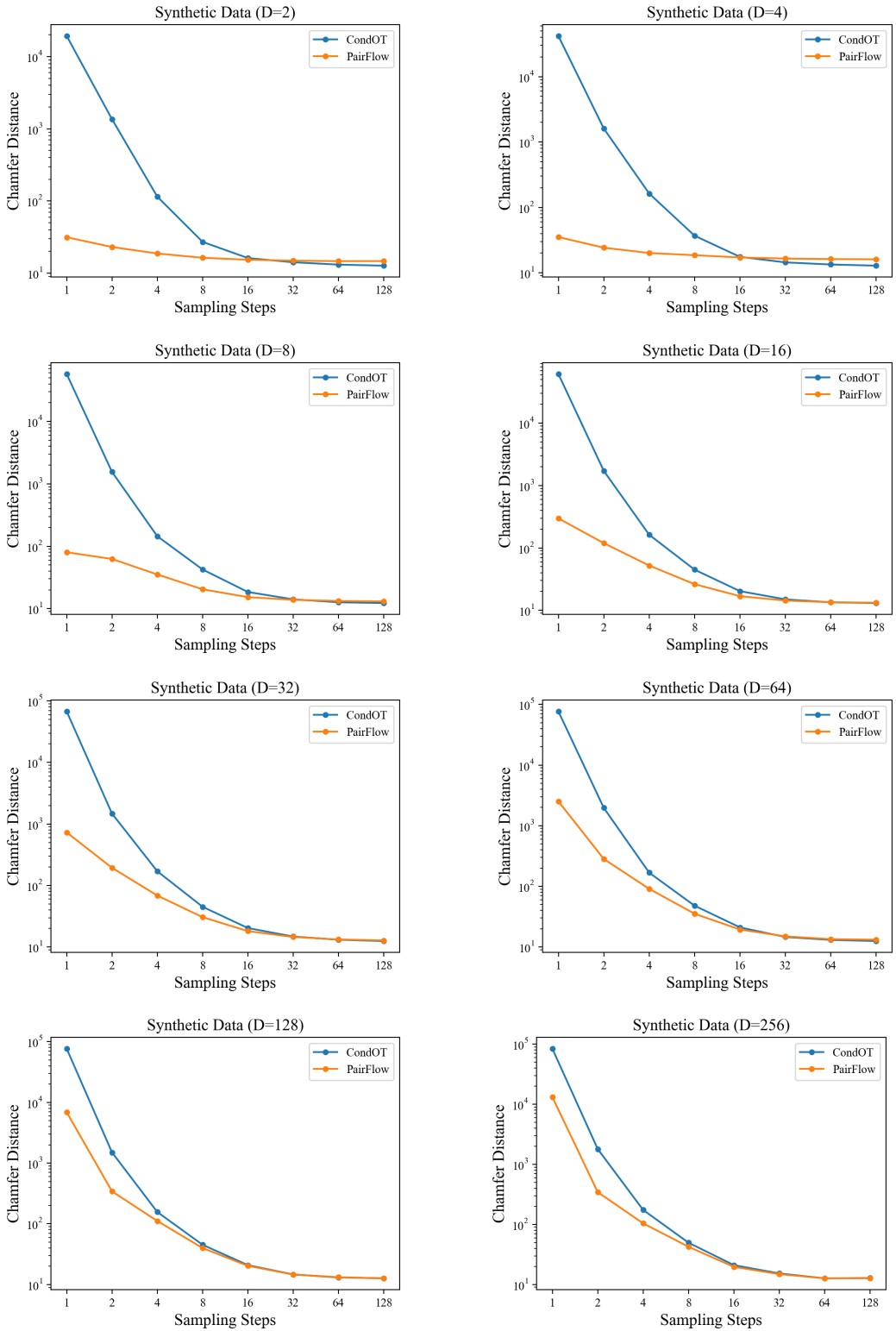

Figure 7: Step-wise performance analysis on synthetic N-fold two-moons. Each plot shows Chamfer distance (log scale) vs. sampling steps for different dimensions $d$. Closed-form pairing (PAIRFLOW) consistently outperforms standard CondOT—especially at few sampling steps—while the margin shrinks as d increases, indicating diminishing gains in high dimensions.

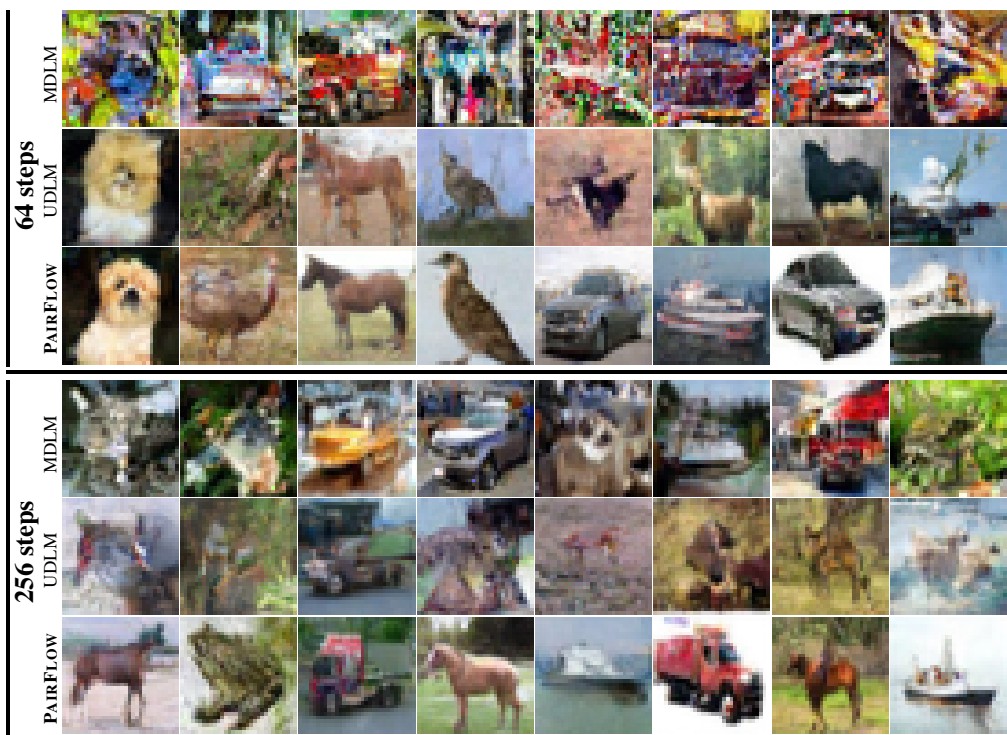

Figure 8: Additional qualitative results of 64-step and 256-step generation on CIFAR-10 ($32 \times 32$).

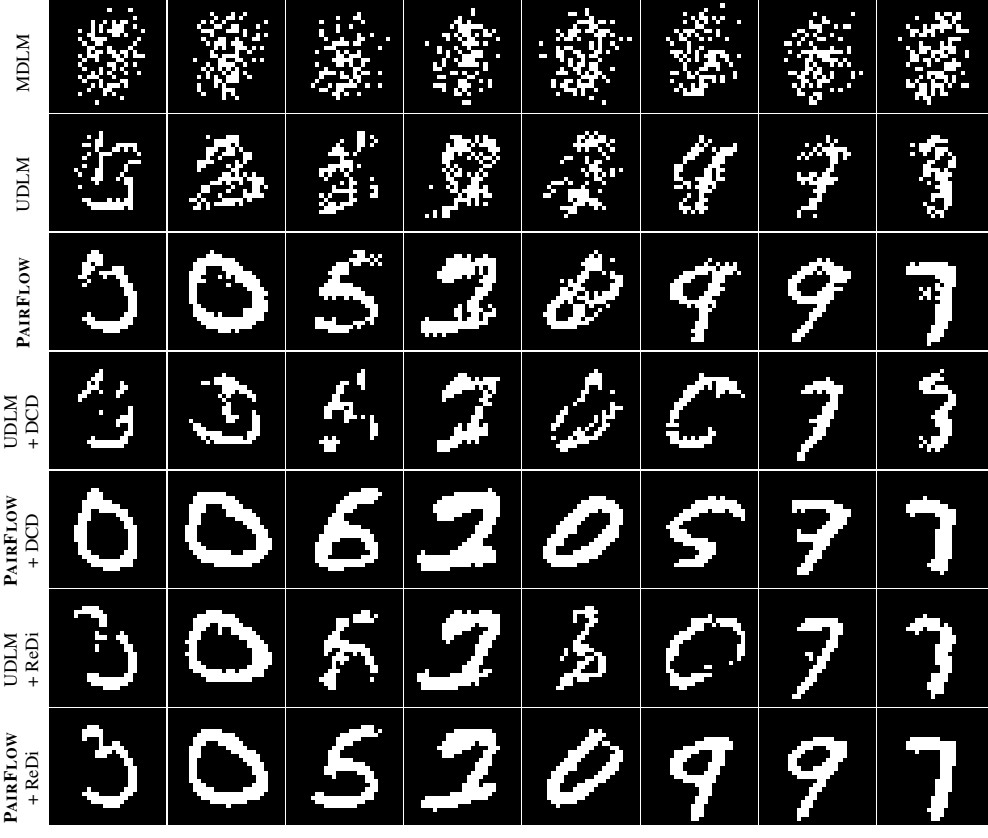

Figure 9: Additional qualitative results of 1-step generation on MNIST-Binary ($28 \times 28$).

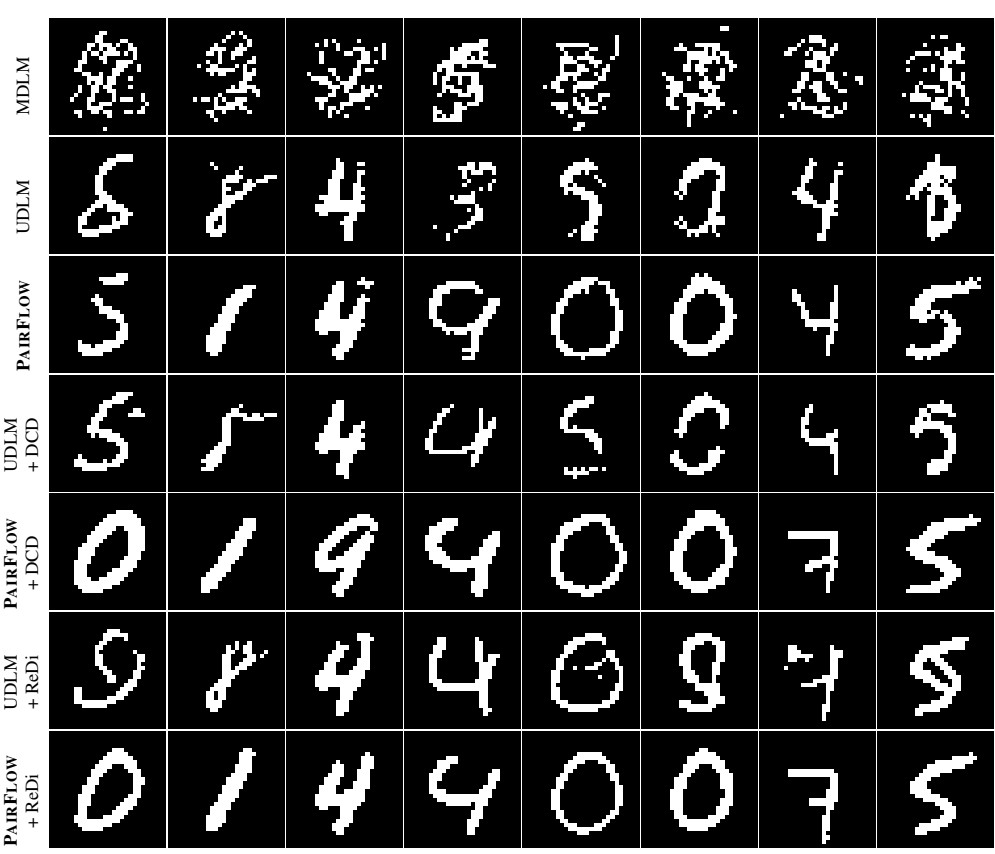

Figure 10: Additional qualitative results of 2-step generation on MNIST-Binary ($28 \times 28$).

