# OpenReview forum: "PairFlow: Closed-Form Source-Target Coupling for Few-Step Generation in Discrete Flow Models"
_ICLR.cc/2026/Conference — ICLR 2026 Poster_

### Official Review · Reviewer_PZ3N · 2025-10-29

**Soundness:** 3
**Presentation:** 2
**Contribution:** 3
**Rating:** 6
**Confidence:** 4

**Summary:**

This paper introduces PAIRFLOW, a lightweight preprocessing method that accelerates sampling in Discrete Flow Models (DFMs) without relying on pretrained teachers or costly finetuning. DFMs, though powerful for modeling discrete data such as molecules and images, typically require many iterative steps for generation. PAIRFLOW addresses this by deriving closed-form forward and backward velocity fields that allow efficient pairing of source (prior) and target (data) samples directly in discrete spaces, in terms of the Hamming distance. This pairing enables the model to learn “straighter” probability paths, drastically reducing the number of sampling steps needed. Despite its minimal computational overhead (≤1.7% of total training cost), PAIRFLOW achieves or surpasses the performance of state-of-the-art distillation-based methods like ReDi and Discrete Consistency Distillation, across both molecular (QM9, ZINC-250k) and image (MNIST-Binary, CIFAR-10) benchmarks. Furthermore, models trained with PAIRFLOW serve as stronger bases for later distillation, achieving additional speed and quality gains, suggesting a general strategy for efficient few-step generation in discrete generative models.

**Strengths:**

- The authors show that the method works well, especially in training small text-to-image models. It can also be combined successfully with distillation methods such as DCD and ReDI.

- Despite the weaknesses I point out below, the method can be a cheap way to boost pre-training of DFMs, and is relevant to the community.

**Weaknesses:**

- The concept of straightness of probability paths is central to the paper, but the explanation in Section 3.2 is lacking and leaves the questions with many questions as to what it means to straighten probability paths. This explains my low presentation score. Happy to raise my score if this is addressed.

- As the authors point out in Section 6, the methods seems to perform better for relatively low dimensional data. This is not surprising, because the construction of the backward trajectories relies on the training dataset having good coverage over the distribution measured in the Hamming distance, which can be hard when the dimension is very high.

- The authors try their method on datasets of tens to low hundreds of thousands of samples. Since evaluating the backward velocity field involves a sum over all training samples, I wonder how scalable the method is when the training set is much larger, which may be needed for good performance in higher dimensional settings (related to the previous point). Can the authors comment on this?

**Questions:**

- Can the proposed method be extended to masked priors? It doesn’t look like it can. Masked priors are relevant because they are the best performing methods for some applications.

- The proposed method is close in spirit to the works in continuous diffusion that use empirical estimates of the score function (with different goals). This kind of approaches are fundamentally cursed by dimension, and although they may provide some gains, are hard and expensive to scale to very high dimensions and very large training datasets. Can the authors comment on this?

---

> ### Author Response · Authors · 2025-11-25
> **Response to reviewer PZ3N (1/3)**
>
> Dear reviewer PZ3N,
>
> We sincerely thank the reviewer for the thoughtful and constructive feedback, and for recognizing the practical relevance of our method, its effectiveness in accelerating discrete flow models, and its compatibility with distillation approaches such as DCD and ReDi. The reviewer’s comments on the straightness concept, scalability, and connections to empirical score estimation were especially valuable and have helped us improve the clarity and positioning of our work. We greatly appreciate the time and care invested in evaluating our paper. We address the questions and concerns below.
>
> ---
>
> ## **[W1] More explanation for the straightness of probability path**
>
> Thank you for highlighting that the explanation of “straightening probability paths” in Section 3.2 was unclear. In short, the straightening probability paths indicates reducing factorization errors. We adopted the term straightness used in the continuous setup since the motivation comes from straightening paths in the continuous domain. We provide a clearer and more detailed explanation below, and **have updated Section 3.2 of the manuscript** accordingly, with revisions highlighted in blue. Any feedback would be appreciated if the following description does not fully address your concern.
>
> ### **[3.2] Straightening Probability Paths for Accelerated Sampling**
>
> The concept of straight probability paths was originally introduced in the continuous domain to enable accelerated sampling. Prior work [1] identified curved probability paths as a key challenge in few-step sampling: when velocity fields are evaluated only at coarse time steps, numerical integration deviates from the true trajectories. liu et al. [1] addressed this issue through ''rectification,'' in which a student flow model is trained on source--target pairs generated by a teacher model, effectively yielding significantly straighter probability paths.
>
> In the discrete setting, this challenge of \textit{path curvature} translates to capturing the \textit{statistical correlations} between tokens. Since DFMs approximate exponentially large joint transitions through fully factorized per-token updates, a mismatch inevitably arises between the true joint transition and its product-form approximation. This discrepancy becomes especially detrimental during few-step generation, where highly correlated tokens must be updated simultaneously. To address this, prior works have primarily relied on distillation-based approaches [2,3,4], aiming to better capture these correlations by explicitly transferring multi-step dependencies from a stronger teacher model.
>
> Yoo et al. [5] formalized this factorization mismatch via conditional Total Correlation (TC), defined as:
>
> $$\text{TC}\_{\pi}(x\_s | x\_t) = \mathbb{E}\_{x\_t} \left[ D\_{\text{KL}} \left( p\_{s|t}(x\_s | x\_t) \ \| \ \prod\_{i=1}^{N} p\_{s|t}(x\_s^i | x\_t) \right) \right]$$
>
> (same as Eqn. [7] of manuscript), which serves as a metric for the factorization error. Crucially, Yoo et al. [5] interpret this factorization error as the discrete analog of path curvature: minimizing TC is equivalent to "straightening" the trajectory by decoupling token transitions. Analogous to ReFlow [1], which rectifies continuous paths, they demonstrate that reducing the conditional Total Correlation requires iteratively refining the source--target coupling $\pi(x_0, x_1)$. To achieve this, they employ an iterative distillation process, alternating between generating improved pairs using the current model and optimizing $\mathcal{L}_{\text{DFM}}$. This procedure effectively finds a "statistically straight" coupling that enables efficient few-step generation.
>
> [1] Flow straight and fast: Learning to generate and transfer data with rectified flow. ICLR 2023.
>
> [2] Beyond Autoregression: Fast LLMs via Self-Distillation Through Time. ICLR 2025.
>
> [3] Distillation of discrete diffusion through dimensional correlations. ICML 2025.
>
> [4] The diffusion duality. ICML 2025.
>
> [5] Redi: Rectified discrete flow. NeurIPS 2025.

---

> ### Author Response · Authors · 2025-11-25
> **Response to reviewer PZ3N (2/3)**
>
> ## **[W2,Q2] Scalability to High-Dimensionality and Large Vocabulary Size**
>
> To evaluate the scalability of our method on more complex and higher-dimensional systems, we additionally tested PairFlow in two substantially larger settings. We decompose complexity into (i) higher dimensionality and (ii) larger vocabulary size with significantly more training data. For (i), we use FFHQ, a 64×64 face image dataset, and for (ii), we use LM1B, a large-scale language modeling corpus with a 30k vocabulary and 3.5M training examples.
>
> ### 1. FFHQ 64x64 (High-dimensional dataset)
>
> For the experiment with FFHQ, the table of FID scores below shows that our method provides consistent improvements, demonstrating that PairFlow remains effective even in higher-dimensional image generation settings. Although the relative gains are smaller than those observed on CIFAR-10, **the improvements remain substantial and remain both consistent and clear.** We added the results of this experiment in Appendix K of the revised manuscript.
>
> $$
> \begin{array}{lccccccccccc}
> \hline
> \text{FID$\downarrow$} & \text{1} & \text{2} & \text{4} & \text{8} & \text{16} & \text{32} & \text{64} & \text{128} & \text{256} & \text{512} & \text{1024} \newline
> \hline
> \text{UDLM} & 403.04 & 399.26 & 363.97 & 273.31 & 153.71 & 97.87 & 74.85 & 63.93 & 59.28 & 55.99 & 55.30 \newline
> \text{PairFlow} & \textbf{394.14} & \textbf{368.36} & \textbf{329.13} & \textbf{243.88} & \textbf{140.05} & \textbf{90.85} & \textbf{69.67} & \textbf{59.86} & \textbf{56.52} & \textbf{54.19} & \textbf{53.18} \newline
> \hline
> \end{array}
> $$
>
> ### 2. LM1B (Large-scale vocabulary and data)
>
> In our experiment with LM1B, we segment the corpus into sequences of varying lengths ($N = 16, 32, 64, 128$) while keeping the total number of training samples fixed ($|X_1| \approx 3.5\text{M}$). Using GPT-2 Large to measure generative perplexity over 1,024 generated samples for each NFE, we find that PairFlow performs reliably across all sequence lengths. As expected, the performance gains decrease as dimensionality increases. However, **PairFlow consistently improves over the baseline at all evaluated scales.** These results indicate that our method remains effective even at the scale of tens of thousands of vocabulary items and millions of data samples. Comprehensive results, along with additional entropy analysis, are provided in Appendix K of the revised manuscript.
>
> $$
> \begin{array}{lccccccccc}
> \hline
> \text{Gen. PPL. $\downarrow$} & \text{4} & \text{8} & \text{16} & \text{32} & \text{64} & \text{128} & \text{256} & \text{512} & \text{1024} \newline
> \hline
> \text{\emph{N=16}} \newline
> \text{UDLM} & 299.18 & 225.90 & 207.17 & 195.80 & 200.77 & \bf{197.04} & 199.12 & \bf{195.37} & 198.22 \newline
> \text{PairFlow} & \bf{242.22} & \bf{208.04} & \bf{200.99} & \bf{190.36} & \bf{191.74} & 199.45 & \bf{188.84} & 196.91 & \bf{198.12} \newline
> \hline
> \text{\emph{N=32}} \newline
> \text{UDLM} & 263.93 & 192.78 & 167.85 & 167.49 & 155.68 & 150.52 & 152.40 & 151.74 & 154.02 \newline
> \text{PairFlow} & \bf{218.48} & \bf{172.27} & \bf{156.35} & \bf{150.53} & \bf{143.83} & \bf{145.77} & \bf{142.54} & \bf{141.04} & \bf{147.57} \newline
> \hline
> \text{\emph{N=64}} \newline
> \text{UDLM} & 214.07 & 150.59 & 130.49 & 120.19 & 117.90 & 116.23 & 112.24 & 113.77 & 115.11 \newline
> \text{PairFlow} & \bf{174.78} & \bf{138.94} & \bf{123.06} & \bf{115.71} & \bf{114.73} & \bf{112.92} & \bf{111.29} & \bf{107.06} & \bf{110.83} \newline
> \hline
> \text{\emph{N=128}} \newline
> \text{UDLM} & 169.61 & 123.48 & 105.13 & 98.94 & 97.89 & 94.92 & 93.75 & 94.12 & 93.59 \newline
> \text{PairFlow} & \bf{167.90} & \bf{121.09} & \bf{102.16} & \bf{96.61} & \bf{93.93} & \bf{91.51} & \bf{90.21} & \bf{89.09} & \bf{89.07} \newline
> \hline
> \end{array}
> $$

---

> ### Author Response · Authors · 2025-11-25
> **Response to reviewer PZ3N (3/3)**
>
> ## **[W3] High Computational Cost on Large-Scale and High-Dimensional Data**
>
> We sincerely thank the reviewer for raising this important point. It is true that deriving the closed-form velocity over the full training set becomes computationally expensive as the dataset grows, and developing more scalable approaches is indeed a promising direction for future research.
>
> To address this concern, we further investigated a practical approximation strategy using subset partitioning. In this additional experiment, the dataset is divided into smaller random subsets, and the velocity is computed exclusively within each subset. The results, including the pairing time cost ($T_{\text{PairFlow}}$), are summarized in the table below. We observe that our subset-based approach significantly reduces computational cost while maintaining performance comparable to full-set pairing, consistently outperforming random pairing. This demonstrates that our method can be effectively scaled with a simple partitioning strategy. Full details and results for the Zinc-250k dataset are provided in Appendix J of the revised manuscript.
>
> $$
> \begin{array}{llccccccc}
> \hline
> \text{Validity$\uparrow$} & \text{$T_{PairFlow}$} & \text{1} & \text{2} & \text{4} & \text{8} & \text{16} & \text{32} & \text{64} \newline
> \hline
> \text{UDLM} & 0 & 0.3{\pm}0.5 & 65.2{\pm}8.2 & 435.7{\pm}14.4 & 775.1{\pm}19.5 & \bf{887.3{\pm}12.7} & \bf{921.5{\pm}8.5} & \bf{937.3{\pm}3.9} \newline
> \text{Random} & 0 & 0.6{\pm}0.9 & 68.3{\pm}10.7 & 351.2{\pm}15.8 & 569.4{\pm}16.6 & 611.0{\pm}16.3 & 602.4{\pm}13.3 & 571.0{\pm}13.2 \newline
> \hline
> \text{PairFlow} \newline
> \text{Full set} & 13\text{m} & 9.9{\pm}2.3 & \bf{146.3{\pm}10.4} & \bf{533.9{\pm}13.9} & 799.4{\pm}9.2 & 873.2{\pm}14.1 & 901.0{\pm}14.2 & 907.8{\pm}7.7 \newline
> \text{2-Subsets} & 6\text{m} & 10.6{\pm}2.8 & 145.7{\pm}13.5 & 530.6{\pm}20.3 & \bf{802.5{\pm}6.9} & 882.6{\pm}7.1 & 902.4{\pm}13.4 & 911.5{\pm}9.2 \newline
> \text{4-Subsets} & 2.9\text{m} & 12.1{\pm}3.3 & 142.5{\pm}5.2 & 509.7{\pm}11.0 & 780.9{\pm}14.4 & 858.9{\pm}7.9 & 886.7{\pm}8.7 & 899.0{\pm}10.5 \newline
> \text{8-Subsets} & 1.5\text{m} & \bf{12.3{\pm}2.5} & 141.1{\pm}6.7 & 510.9{\pm}17.2 & 766.8{\pm}12.6 & 857.5{\pm}8.4 & 886.9{\pm}9.6 & 896.5{\pm}6.3 \newline
> \hline
> \end{array}
> $$
>
> ---
>
> ## **[Q1] Applicability to Masked Source Distributions**
>
> You are right. For masked priors, the source distribution collapses to a single deterministic state, causing all flows to originate from the same point. In this setting, defining a meaningful source–target coupling becomes impossible, which further supports our decision to build our method on the uniform distribution.
>
> While masked discrete flow matching provides the best performance for some applications, we note that, particularly for few-step generation, uniform priors have advantages over masked priors, as well documented in prior work and consistently reflected in our experiments.

---

### Official Review · Reviewer_vdPJ · 2025-11-01

**Soundness:** 3
**Presentation:** 3
**Contribution:** 3
**Rating:** 6
**Confidence:** 2

**Summary:**

This paper presents PairFlow, a novel and highly efficient preprocessing method for training Discrete Flow Models (DFMs) that enables high-quality generation with very few sampling steps. Inspired by ReFlow and ReDi, the authors uses coupled source and target distributions for training. The core innovation is the derivation of closed-form forward and backward velocity fields for DFMs, which allows for the direct construction of optimized source-target data pairs from a collection of samples from the target distribution, without needing a pretrained teacher model.

**Strengths:**

1. The paper is well-motivated and easy to follow.
2. The derivation of the closed-form velocity field of DFM is a significant theoretical contribution.
3. The proposed method has a huge improvement in computation complexity.

**Weaknesses:**

1. The continuous flow experiments in Appendix E.3 suggest that the advantage of closed-form pairing may diminish with increasing data dimensionality. A brief discussion and further experiments on the scalability of the discrete PairFlow method to very large vocabularies and sequence lengths (Image datasets of higher resolution or language modeling) would be beneficial for setting expectations for future applications.
2. Typos in Line 127-128: the codomain of $p_t(\cdot)$ and $v_t(\cdot)$.

**Questions:**

PairFlow now focus on uniform source distribution. What's the difficulties of (or reasons of not) considering a more general class of source distributions?

---

> ### Author Response · Authors · 2025-11-25
> **Response to reviewer vdPJ**
>
> Dear reviewer vdPJ,
>
> We sincerely thank the reviewer for the thoughtful and encouraging feedback, and for recognizing the motivation, clarity, and theoretical contributions of our work—particularly the closed-form velocity derivation and the computational efficiency of PairFlow. The reviewer’s comments on scalability and broader applicability were highly valuable and have helped strengthen the paper. We also thank the reviewer for pointing out the typo, which we have corrected and highlighted in the updated PDF. We address the questions and concerns below.
>
> ---
>
> ## **[W1] Scalability to High-Dimensionality and Large Vocabulary Size**
>
> To evaluate the scalability of our method on more complex and higher-dimensional systems, we additionally tested PairFlow in two substantially larger settings. We decompose complexity into (i) higher dimensionality and (ii) larger vocabulary size with significantly more training data. For (i), we use FFHQ, a 64×64 face image dataset, and for (ii), we use LM1B, a large-scale language modeling corpus with a 30k vocabulary and 3.5M training examples.
>
> ### 1. FFHQ 64x64 (High-dimensional dataset)
>
> For the experiment with FFHQ, the table of FID scores below shows that our method provides consistent improvements, demonstrating that PairFlow remains effective even in higher-dimensional image generation settings. Although the relative gains are smaller than those observed on CIFAR-10, **the improvements remain substantial and remain both consistent and clear.** We added the results of this experiment in Appendix K of the revised manuscript.
>
> $$
> \begin{array}{lccccccccccc}
> \hline
> \text{FID}\downarrow&\text{1}&\text{2}&\text{4}&\text{8}&\text{16}&\text{32}&\text{64}&\text{128}&\text{256}&\text{512}&\text{1024}\\\\
> \hline
> \text{UDLM}&403.04&399.26&363.97&273.31&153.71&97.87&74.85&63.93&59.28&55.99&55.30\\\\
> \text{PairFlow}&\textbf{394.14}&\textbf{368.36}&\textbf{329.13}&\textbf{243.88}&\textbf{140.05}&\textbf{90.85}&\textbf{69.67}&\textbf{59.86}&\textbf{56.52}&\textbf{54.19}&\textbf{53.18}\\\\
> \hline
> \end{array}
> $$
>
> ### 2. LM1B (Large-scale vocabulary and data)
>
> In our experiment with LM1B, we segment the corpus into sequences of varying lengths ($N=16,32,64,128$) while keeping the total number of training samples fixed ($|X_1|\approx 3.5\text{M}$). Using GPT-2 Large to measure generative perplexity over 1,024 generated samples for each NFE, we find that PairFlow performs reliably across all sequence lengths. As expected, the performance gains decrease as dimensionality increases. However, **PairFlow consistently improves over the baseline at all evaluated scales.** These results indicate that our method remains effective even at the scale of tens of thousands of vocabulary items and millions of data samples. Comprehensive results, along with additional entropy analysis, are provided in Appendix K of the revised manuscript.
>
> $$
> \begin{array}{lccccccccc}
> \hline
> \text{Gen. PPL.}\downarrow&\text{4}&\text{8}&\text{16}&\text{32}&\text{64}&\text{128}&\text{256}&\text{512}&\text{1024}\\\\
> \hline
> \text{\emph{N=16}}\\\\
> \text{UDLM}&299.18&225.90&207.17&195.80&200.77&\bf{197.04}&199.12&\bf{195.37}&198.22\\\\
> \text{PairFlow}&\bf{242.22}&\bf{208.04}&\bf{200.99}&\bf{190.36}&\bf{191.74}&199.45&\bf{188.84}&196.91&\bf{198.12}\\\\
> \hline
> \text{\emph{N=32}}\\\\
> \text{UDLM}&263.93&192.78&167.85&167.49&155.68&150.52&152.40&151.74&154.02\\\\
> \text{PairFlow}&\bf{218.48}&\bf{172.27}&\bf{156.35}&\bf{150.53}&\bf{143.83}&\bf{145.77}&\bf{142.54}&\bf{141.04}&\bf{147.57}\\\\
> \hline
> \text{\emph{N=64}}\\\\
> \text{UDLM}&214.07&150.59&130.49&120.19&117.90&116.23&112.24&113.77&115.11\\\\
> \text{PairFlow}&\bf{174.78}&\bf{138.94}&\bf{123.06}&\bf{115.71}&\bf{114.73}&\bf{112.92}&\bf{111.29}&\bf{107.06}&\bf{110.83}\\\\
> \hline
> \text{\emph{N=128}}\\\\
> \text{UDLM}&169.61&123.48&105.13&98.94&97.89&94.92&93.75&94.12&93.59\\\\
> \text{PairFlow}&\bf{167.90}&\bf{121.09}&\bf{102.16}&\bf{96.61}&\bf{93.93}&\bf{91.51}&\bf{90.21}&\bf{89.09}&\bf{89.07}\\\\
> \hline
> \end{array}
> $$
>
> ---
>
> ## **[W2] Correction of Codomain Notation**
>
> Thank you for pointing out. We have corrected the typo and marked the change in blue in the revised manuscript.
>
> - $p_t(x):\mathcal{V}^N\times[0,1]\to\textcolor{blue}{[0,1]}$
> - $v_t(x):\mathcal{V}^N\times[0,1]\to\textcolor{blue}{\mathbb{R}^{N\times K}}$
>
> ---
>
> ## **[Q1] Considering other source distribution**
>
> Discrete flow matching is typically modeled with two types of source distributions: masked and uniform. The mask source distribution collapses to a single deterministic state, causing all flows to originate from the same point. In this setting, defining a meaningful source–target coupling becomes impossible, which further supports our decision to build our method on the uniform distribution.
>
> We also note that, for few-step generation, uniform source distribution is known to perform better than mask source distribution, a behavior that is well documented in prior work and consistently reflected in our experiments.

---

### Official Review · Reviewer_au4X · 2025-11-03

**Soundness:** 3
**Presentation:** 3
**Contribution:** 2
**Rating:** 6
**Confidence:** 3

**Summary:**

PairFlow introduces a pre-processing step for training discrete flow models to enable few-step sampling without the requirement of a pre-trained teacher model. The approach achieves this through around a closed-form inversion step for discrete flow models, enabling the development of source-target pairs which can be used training PairFlow.

**Strengths:**

The approach is mathematically interesting, and offers significant practical utility, as it eliminates the need for a pre-trained teacher for distillation while accelerating inference through few-step sampling in discrete settings.

**Weaknesses:**

Deriving the closed-form velocity field although possible requires summing over all the training data, which can become significant for large systems. Further, it is possible that using the analytical vector field to determine source-target pairs leads to a distilled model that overfits to the data due to the way the analytical field is obtained. It would be useful to demonstrate that this in fact does not occur. There is some evidence to suggest that this may be happening (novelty of molecules on the molecular datasets is the worst compared to other baselines).

**Questions:**

- Using the closed-form velocity field to generate source-target pairs may lead to the distilled model being more likely to overfit to the training data. This has been observed in continuous settings by Bertrand et al. (2025). Proving that this isn't the case would be useful across the considered datasets (my concerns around the low novelty for QM9 validate these concerns).
- Do additional Reflow steps (deriving a new analytical vector field on reflow samples from each distilled model) applied to subsequent trained models improve performance across metrics? In other words, does the concept of straightening the paths still apply in the discrete setting with multiple iterations of distilled models?
- How does the approach perform on more complex systems, like ImageNet? Since it's observed that the continuous approach fails for higher-dimensional systems, does this also hold for the discrete setting?
- How does the approach perform compared to continuous flow matching methods? A contrast on the already-shown image baselines, e.g., MNIST, CIFAR10, etc., would be useful.

---

> ### Author Response · Authors · 2025-11-25
> **Response to reviewer au4X (1/3)**
>
> Dear reviewer au4X,
>
> We sincerely thank the reviewer for the thoughtful and constructive feedback, as well as for recognizing both the mathematical interest and the practical utility of our proposed PairFlow framework. Your comments on overfitting, scalability, and comparisons to continuous flow matching were especially valuable, and have helped us further strengthen the clarity and rigor of our paper. We address the questions and concerns below.
>
> ---
>
> ## **[W1] Calculate pairs for large dataset is computationally heavy**
>
> We sincerely thank the reviewer for raising this important point. It is true that deriving the closed-form velocity over the full training set becomes computationally expensive as the dataset grows, and developing more scalable approaches is indeed a promising direction for future research.
>
> To address this concern, we further investigated a practical approximation strategy using subset partitioning. In this additional experiment, the dataset is divided into smaller random subsets, and the velocity is computed exclusively within each subset. The results, including the pairing time cost ($T_{\text{PairFlow}}$), are summarized in the table below. We observe that our subset-based approach significantly reduces computational cost while maintaining performance comparable to full-set pairing, consistently outperforming random pairing. This demonstrates that our method can be effectively scaled with a simple partitioning strategy. Full details and results for the Zinc-250k dataset are provided in Appendix J of the revised manuscript.
>
> $$
> \begin{array}{llccccccc}
> \hline
> \text{Validity$\uparrow$}&\text{$T_{PairFlow}$}&\text{1}&\text{2}&\text{4}&\text{8}&\text{16}&\text{32}&\text{64} \\\\
> \hline
> \text{UDLM} & 0 & 0.3{\pm}0.5 & 65.2{\pm}8.2 & 435.7{\pm}14.4 & 775.1{\pm}19.5 & \bf{887.3{\pm}12.7} & \bf{921.5{\pm}8.5} & \bf{937.3{\pm}3.9} \\\\
> \text{Random} & 0 & 0.6{\pm}0.9 & 68.3{\pm}10.7 & 351.2{\pm}15.8 & 569.4{\pm}16.6 & 611.0{\pm}16.3 & 602.4{\pm}13.3 & 571.0{\pm}13.2 \\\\
> \hline
> \text{PairFlow} \\\\
> \text{Full set} & 13\text{m} & 9.9{\pm}2.3 & \bf{146.3{\pm}10.4} & \bf{533.9{\pm}13.9} & 799.4{\pm}9.2 & 873.2{\pm}14.1 & 901.0{\pm}14.2 & 907.8{\pm}7.7 \\\\
> \text{2-Subsets} & 6\text{m} & 10.6{\pm}2.8 & 145.7{\pm}13.5 & 530.6{\pm}20.3 & \bf{802.5{\pm}6.9} & 882.6{\pm}7.1 & 902.4{\pm}13.4 & 911.5{\pm}9.2 \\\\
> \text{4-Subsets} & 2.9\text{m} & 12.1{\pm}3.3 & 142.5{\pm}5.2 & 509.7{\pm}11.0 & 780.9{\pm}14.4 & 858.9{\pm}7.9 & 886.7{\pm}8.7&899.0{\pm}10.5 \\\\
> \text{8-Subsets} & 1.5\text{m} & \bf{12.3{\pm}2.5} & 141.1{\pm}6.7 & 510.9{\pm}17.2 & 766.8{\pm}12.6 & 857.5{\pm}8.4 & 886.9{\pm}9.6 & 896.5{\pm}6.3 \\\\
> \hline
> \end{array}
> $$

---

> ### Author Response · Authors · 2025-11-25
> **Response to reviewer au4X (2/3)**
>
> ## **[W2,Q1] Analytical vector field can cause overfit to the training set**
>
> Although novelty decreases slightly in the QM9 experiments, this trend does not appear in the other experiments. On Zinc-250k, the novelty trend of PairFlow closely matches that of the UDLM baseline, indicating that our method does not exhibit additional overfitting. Also for image generation, we additionally computed DINO-v2–based FID and Inception-based FID scores on the test set and nearest-neighbor (NN) distances to the training set on CIFAR-10, following Bertrand et al. (2025). The DINO-FID results follow the same trend as in our main paper, and the NN distances are nearly identical to those of the baseline. We have added these results in Appendix L of the revised manuscript. The overall results demonstrate that our method does not suffer from overfitting in most cases.
>
>
> $$
> \begin{array}{lccccccccccc}
> \hline
> \text{FID$\downarrow$}&\text{1}&\text{2}&\text{4}&\text{8}&\text{16}&\text{32}&\text{64}&\text{128}&\text{256}&\text{512}&\text{1024}\\\\
> \hline
> \text{UDLM}&306.45&296.77&266.64&178.11&114.04&80.40&62.70&53.83&50.96&47.61&47.16\\\\
> \text{PairFlow}&\bf{235.65}&\bf{247.18}&\bf{209.94}&\bf{137.16}&\bf{94.24}&\bf{67.53}&\bf{51.48}&\bf{43.79}&\bf{42.44}&\bf{40.85}&\bf{39.92}\\\\
> \hline
> \text{FID-Dino$\downarrow$}&\text{1}&\text{2}&\text{4}&\text{8}&\text{16}&\text{32}&\text{64}&\text{128}&\text{256}&\text{512}&\text{1024}\\\\
> \hline
> \text{UDLM}&2448.46&2410.65&1975.58&1344.44&959.39&755.80&646.47&598.53&598.54&553.17&560.75\\\\
> \text{PairFlow}&\bf{2059.26}&\bf{1988.21}&\bf{1626.08}&\bf{1127.30}&\bf{828.09}&\bf{623.89}&\bf{530.70}&\bf{486.59}&\bf{484.27}&\bf{470.33}&\bf{470.05}\\\\
> \hline
> \end{array}
> $$
>
> $$
> \begin{array}{lccccccccccc}
> \hline
> \text{L2$\uparrow$}&\text{1}&\text{2}&\text{4}&\text{8}&\text{16}&\text{32}&\text{64}&\text{128}&\text{256}&\text{512}&\text{1024}\\\\
> \hline
> \text{UDLM}&7.97&\bf{9.13}&\bf{10.06}&\bf{10.06}&\bf{9.40}&\bf{8.94}&\bf{8.63}&8.41&8.29&8.29&8.22\\\\
> \text{PairFlow}&\bf{8.03}&8.76&9.42&9.56&9.18&8.75&8.63&\bf{8.55}&\bf{8.51}&\bf{8.52}&\bf{8.52}\\\\
> \hline
> \text{Cosine(Dino)$\downarrow$}&\text{1}&\text{2}&\text{4}&\text{8}&\text{16}&\text{32}&\text{64}&\text{128}&\text{256}&\text{512}&\text{1024}\\\\
> \hline
> \text{UDLM}&\bf{0.242}&\bf{0.227}&\bf{0.231}&0.241&0.237&0.238&0.235&0.237&0.237&\bf{0.235}&0.237\\\\
> \text{PairFlow}&0.245&0.228&0.235&\bf{0.239}&\bf{0.236}&\bf{0.232}&\bf{0.232}&\bf{0.230}&\bf{0.232}&\bf{0.235}&\bf{0.233}\\\\
> \hline
> \end{array}
> $$
>
> ---
>
>
> ## **[Q2] Do additional re-flow steps can consistently make improvement in DFM?**
>
> Yes, we find that **additional ReFlow steps consistently improve performance.** On QM9, both UDLM and PairFlow benefit from 1–3 ReFlow iterations, with each step providing incremental gains—especially in the few-step generation regime. We provide the results for 1, 2, and 3 ReFlow steps in the table below. This aligns with prior observations in Yoo et al. (2025), where iterative ReFlow further straightens the flow; our results confirm that the same effect extends to the discrete setting. Moreover, PairFlow consistently outperforms the baseline—not only at the same iteration level but **even when comparing UDLM with multiple ReFlow iterations against PairFlow alone or with only a single additional iteration.** We added full results of this experiment for the QM9 dataset in Appendix I of the revised manuscript. We appreciate the reviewer’s insightful suggestion.
>
> $$
> \begin{array}{lccccccc}
> \hline
> \text{Validity$\uparrow$}&\text{1}&\text{2}&\text{4}&\text{8}&\text{16}&\text{32}&\text{64}\\\\
> \hline
> \text{UDLM}&17.5{\pm}3.2&125.5{\pm}11.8&497.6{\pm}8.3&826.6{\pm}10.3&953.5{\pm}6.1&\bf{991.9{\pm}4.2}&\underline{1000.1{\pm}3.5}\\\\
> \text{+Re-Flow}&59.7{\pm}8.8&232.4{\pm}9.2&588.4{\pm}15.8&849.6{\pm}14.2&940.5{\pm}8.5&967.5{\pm}5.2&978.8{\pm}5.2\\\\
> \text{+Re-Flow}&160.9{\pm}11.8&368.0{\pm}11.8&673.3{\pm}11.2&878.5{\pm}11.6&945.4{\pm}8.1&967.9{\pm}7.4&975.8{\pm}8.0\\\\
> \text{+Re-Flow}&280.2{\pm}6.7&470.7{\pm}18.6&742.2{\pm}18.7&897.9{\pm}9.4&945.3{\pm}5.2&965.0{\pm}5.9&972.6{\pm}7.1\\\\
> \hline
> \text{PairFlow}&223.4{\pm}12.7&416.0{\pm}12.4&734.9{\pm}7.2&921.5{\pm}11.0&\bf{977.1{\pm}3.9}&\underline{990.9{\pm}5.9}&\bf{1000.2{\pm}4.5}\\\\
> \text{+Re-Flow}&361.0{\pm}115.2&512.6{\pm}44.2&775.7{\pm}10.0&929.1{\pm}11.6&\underline{976.2{\pm}4.5}&985.6{\pm}6.7&993.1{\pm}7.1\\\\
> \text{+Re-Flow}&\underline{443.4{\pm}13.4}&\underline{598.6{\pm}18.4}&\underline{823.0{\pm}16.0}&\underline{935.2{\pm}7.7}&969.0{\pm}5.7&984.1{\pm}6.0&989.0{\pm}3.8\\\\
> \text{+Re-Flow}&\bf{529.2{\pm}10.6}&\bf{688.0{\pm}11.3}&\bf{863.6{\pm}9.5}&\bf{945.9{\pm}4.5}&975.7{\pm}6.1&982.6{\pm}7.1&990.2{\pm}7.2\\\\
> \hline
> \end{array}
> $$
>
> ---

---

> ### Author Response · Authors · 2025-11-25
> **Response to reviewer au4X (3/3)**
>
> ## **[Q3] Effectiveness in More Complex System**
>
> To evaluate the scalability of our method on more complex and higher-dimensional systems, we additionally tested PairFlow in two substantially larger settings. Due to computational resource constraints, we decompose complexity into (i) higher dimensionality and (ii) larger vocabulary size with significantly more training data. For (i), we use FFHQ, a 64×64 face image dataset, and for (ii), we use LM1B, a large-scale language modeling corpus with a 30k vocabulary and 3.5M training examples.
>
> ### 1. FFHQ 64x64 (High-dimensional dataset)
>
> For the experiment with FFHQ, the table of FID scores below shows that our method provides consistent improvements, demonstrating that PairFlow remains effective even in higher-dimensional image generation settings. Although the relative gains are smaller than those observed on CIFAR-10, **the improvements remain substantial and remain both consistent and clear.** We added the results of this experiment in Appendix K of the revised manuscript.
>
> $$
> \begin{array}{lccccccccccc}
> \hline
> \text{FID$\downarrow$} & \text{1} & \text{2} & \text{4} & \text{8} & \text{16} & \text{32} & \text{64} & \text{128} & \text{256} & \text{512} & \text{1024} \\\\
> \hline
> \text{UDLM} & 403.04 & 399.26 & 363.97 & 273.31 & 153.71 & 97.87 & 74.85 & 63.93 & 59.28 & 55.99 & 55.30 \\\\
> \text{PairFlow} & \textbf{394.14} & \textbf{368.36} & \textbf{329.13} & \textbf{243.88} & \textbf{140.05} & \textbf{90.85} & \textbf{69.67} & \textbf{59.86} & \textbf{56.52} & \textbf{54.19} & \textbf{53.18} \\\\
> \hline
> \end{array}
> $$
>
> ### 2. LM1B (Large-scale vocabulary and data)
>
> In our experiment with LM1B, we segment the corpus into sequences of varying lengths ($N = 16, 32, 64, 128$) while keeping the total number of training samples fixed ($|X_1| \approx 3.5\text{M}$). Using GPT-2 Large to measure generative perplexity over 1,024 generated samples for each NFE, we find that PairFlow performs reliably across all sequence lengths. As expected, the performance gains decrease as dimensionality increases. However, **PairFlow consistently improves over the baseline at all evaluated scales.** These results indicate that our method remains effective even at the scale of tens of thousands of vocabulary items and millions of data samples. Comprehensive results, along with additional entropy analysis, are provided in Appendix K of the revised manuscript.
>
> $$
> \begin{array}{lccccccccc}
> \hline
> \text{Gen. PPL. $\downarrow$} & \text{4} & \text{8} & \text{16} & \text{32} & \text{64} & \text{128} & \text{256} & \text{512} & \text{1024} \\\\
> \hline
> \text{\emph{N=16}} \\\\
> \text{UDLM} & 299.18 & 225.90 & 207.17 & 195.80 & 200.77 & \bf{197.04} & 199.12 & \bf{195.37} & 198.22 \\\\
> \text{PairFlow} & \bf{242.22} & \bf{208.04} & \bf{200.99} & \bf{190.36} & \bf{191.74} & 199.45 & \bf{188.84} & 196.91 & \bf{198.12} \\\\
> \hline
> \text{\emph{N=32}} \\\\
> \text{UDLM} & 263.93 & 192.78 & 167.85 & 167.49 & 155.68 & 150.52 & 152.40 & 151.74 & 154.02 \\\\
> \text{PairFlow} & \bf{218.48} & \bf{172.27} & \bf{156.35} & \bf{150.53} & \bf{143.83} & \bf{145.77} & \bf{142.54} & \bf{141.04} & \bf{147.57} \\\\
> \hline
> \text{\emph{N=64}} \\\\
> \text{UDLM} & 214.07 & 150.59 & 130.49 & 120.19 & 117.90 & 116.23 & 112.24 & 113.77 & 115.11 \\\\
> \text{PairFlow} & \bf{174.78} & \bf{138.94} & \bf{123.06} & \bf{115.71} & \bf{114.73} & \bf{112.92} & \bf{111.29} & \bf{107.06} & \bf{110.83} \\\\
> \hline
> \text{\emph{N=128}} \\\\
> \text{UDLM} & 169.61 & 123.48 & 105.13 & 98.94 & 97.89 & 94.92 & 93.75 & 94.12 & 93.59 \\\\
> \text{PairFlow} & \bf{167.90} & \bf{121.09} & \bf{102.16} & \bf{96.61} & \bf{93.93} & \bf{91.51} & \bf{90.21} & \bf{89.09} & \bf{89.07} \\\\
> \hline
> \end{array}
> $$
>
> ---
>
> ## **[Q4] Compare the result with continuous flow models**
>
> In Appendix E.2 and Table 14, we compare the base continuous flow matching model with our PairFlow applied on top of it on the MNIST dataset. The results show clear improvements particularly at lower NFEs, indicating that our method is applicable to both discrete and continuous flow models.

---

### Official Review · Reviewer_Vm4B · 2025-11-08

**Soundness:** 2
**Presentation:** 3
**Contribution:** 4
**Rating:** 6
**Confidence:** 5

**Summary:**

This paper proposes a new training method to accelerate discrete flow matching sampling, achieving good results on various tasks, especially CIFAR-10 generation.

The method, called PairFlow, is conceptually similar to ReFlow (Rectified Flow), which is commonly used in the continuous domain. However, PairFlow differs from ReFlow in that it starts from the true data domain and moves backward toward the noise domain using a backward velocity field.

In general, this paper aims to generalize the results obtained in the continuous domain—such as the closed-form velocity formulation and the ReFlow training approach—back to the discrete domain.

**Strengths:**

- Starting from real data, back to noise. This is just a little bit change of the reflow. But I think it is a cool idea. Intuitively, if your reflow model is not trained or sampled well, then the quality of your sampled-images maybe much worse than that of real data. But starting from clean image, you can avoid this.
- this paper has some smart ideas to get the theoretical results, for example, how to get the closed-form formula of forward velocity (A.1). Although the proof about the closed-formula of backward velocity has some flaws, but the direction of that proof (A.2) I think it's correct.

**Weaknesses:**

The main weakness of this paper is dued to its proof of the closed-form formular of the backward velocity.

- line 770 ~ line 792, has so many typos. Those typos make the proof unreadable, though although I can grasp the approach the authors intended to use.
- I do some calculations, and find some part results are right, but the proof process is flawed. There are numerous algebraic mistakes (e.g., signs flipped, missing ±1 terms, and other careless errors) that makes people suspect the correctness of the results.

**Questions:**

Could you please check and rewrite the proof in Appendix A.2? I think the overall approach is sound, but there are too many careless errors. If you can make it accurate and typo-free, I will raise my score.

**Details Of Ethics Concerns:**

no concerns.

---

> ### Author Response · Authors · 2025-11-25
> **Response to Reviewer Vm4B**
>
> Dear reviewer Vm4B
>
> We thank the reviewer for the insightful feedback and for recognizing the core theoretical contributions of our work, particularly the closed-form velocity formulations for discrete flow matching. Your comments greatly helped us strengthen and enrich the paper, and we address the Appendix A.2 proof-related concerns in the response below.
>
> ---
>
> ## **[W1] Typos and flaw of the proof of backward velocity**
>
> We would like to sincerely thank the reviewer for carefully identifying the issues in our earlier version of Appendix A.2. Your detailed comments were extremely helpful. We have thoroughly revised the full derivation, corrected every problematic step, and updated our submission with all modifications clearly highlighted in blue.
>
> In detail, the issue originated from the way the four cases were decomposed in the earlier version of the derivation. The distinction between the $ j = i $ term and the $ j \neq i $ terms should have been handled independently. We also recognized some typos in L770–L792. Please check out the detailed list of modifications below. Please note that, after revising the proof, we confirmed that all results remain unchanged.
>
> ---
>
> - **Add Eqn. [45,47–49]:** To clarify the independence between the $ j = i $ and $ j \neq i $ components.
> - **L775, L788:** Explicitly indicate the target dimension to be separated into four cases.
> - **L797, L810:** Provide clearer descriptions for **Case 2** and **Case 4**.
> - **Fix typos:** Eqn. [50, 52, 54] – Remove the $-k$ term from the superscript of the $ \sum $.
> - **Fix typos:** Eqn. [51, 53, 55] – Correct the sign of the $ \pm 1 $ term in the exponents (add / remove / flip as needed).
> - **Fix typos:** L790, L805 – Change the $+$ symbol to a subscript.
> - **L814, L842:** Add a clear explanation of how Eqn. [56] and Eqn. [62] are derived.
>
> ---
>
> Additionally, to further enhance readability, we reorganized the proof into a more streamlined and intuitive form and included this improved presentation in Appendix H. Importantly, we confirm that both the original formulation and this clearer exposition lead to the same conclusion we initially established.

---

### Public Comment · ~Zhangzhi_Peng1 · 2026-03-12
**is the code available?**

is the code available?

---

### Meta-Review · Area_Chair_NvhY · 2026-01-07

**Summary:**

This paper introduces PairFlow, a lightweight preprocessing method for Discrete Flow Models (DFMs), improving few-step sampling without requiring pretrained teacher models. The method provides significant computational efficiency and enhances generative performance on molecular and image datasets.

**Reviewer Concerns:**

The authors have effectively addressed several key concerns raised by the reviewers. They provided a practical solution for scalability by introducing a subset partitioning strategy, which reduces computational cost for large datasets and demonstrated its effectiveness on the Zinc-250k dataset. They also successfully addressed concerns about overfitting, providing additional experimental results that show PairFlow does not overfit, particularly on QM9 and CIFAR-10. Additionally, they clarified the concept of straightening probability paths in Section 3.2, offering a more detailed explanation that links it to reducing factorization errors and improving token correlations, addressing the main issues with the presentation.

However, some concerns remain outstanding. Although the authors demonstrated scalability on datasets like FFHQ and LM1B, further experiments on larger, high-dimensional datasets, such as ImageNet, are necessary to fully assess the method's scalability. Additionally, while the authors clarified the novelty of applying continuous flow techniques to the discrete setting, the novelty remains somewhat limited, as the core concepts closely follow existing continuous models. Finally, while the paper shows promising results on moderate-scale systems, further validation on complex, real-world datasets is needed to strengthen the paper's claims about the generalizability of PairFlow.

BTW, there is a potential hallucinated reference with a completely wrong set of authors.

Ciprian Chelba, Tomas Mikolov, Mike Schuster, Qi Ge, Thorsten Brants, Phillipp Koehn, and Tony Robinson. Exploring the limits of language modeling. arXiv preprint arXiv:1312.3005, 2013.

**Reviewer Scores:**

Not sure if Reviewer vdPJ would increase the score, but the other three reviewers are likely to do so given that most of their concerns have been adequately addressed.

---

### Decision · Program_Chairs · 2026-01-26

Accept (Poster)